# HIGHWAY REINFORCEMENT LEARNING

## ABSTRACT

Traditional Dynamic Programming (DP) approaches suffer from slow backward credit-assignment (CA): one time step per update. A popular solution for multi-step CA is to use multi-step Bellman operators. Existing control methods, however, typically suffer from large variance of multi-step off-policy corrections or are biased, preventing convergence. To overcome these problems, we introduce a novel multi-step Bellman Optimality operator, which quickly transports credit information from the future into the past through multiple "highways" induced by various behavioral policies. Our operator is unbiased with respect to the optimal value function and converges faster than the traditional Bellman Optimality operator. Its computational complexity is linear in the number of behavioral policies and lookahead depth. Moreover, it yields a family of novel multi-step off-policy algorithms that do not need importance sampling. We derive a convergent multi-step off-policy variant of Q-learning called Highway Q-Learning, and also a deep function approximation variant called Highway DQN. Experiments on toy tasks and visual MinAtar Games (Young & Tian, 2019) illustrate that our algorithms outperform similar multi-step methods.

## 1 INTRODUCTION

Recent advances in multi-step reinforcement learning (RL) have achieved remarkable empirical success (Horgan et al., 2018; Barth-Maron et al., 2018). However, a major challenge of multi-step RL is to balance the trade-off between traditional "safe" one-time-step-per-trial credit assignment (CA) relying on knowledge stored in a learned Value Function (VF), and large CA jumps across many time steps. A traditional way of addressing this issue is to impose a fixed prior distribution over the possible numbers of CA steps, e.g., TD($\lambda$) (Sutton & Barto, 2018), GAE($\lambda$) (Schulman et al., 2016). This typically ignores the current state-specific quality of the current VF, which dynamically improves during learning. Besides, the prior distribution usually has to be tuned case by case.

Multi-step RL should also work for off-policy learning, that is, learning from data obtained by other behavioral policies. Most previous research on this has focused on Policy Iteration(PI)-based approaches (Sutton & Barto, 2018), which need to correct the discrepancy between target policy and behavior policy to evaluate the VF (Precup, 2000; Harutyunyan et al., 2016; Munos et al., 2016; Sutton & Barto, 2018; Schulman et al., 2016). Classic importance sampling(IS)-based methods is proven to be unbiased, but suffer from high variance due to the product of IS ratios (Cortes et al., 2010; Metelli et al., 2018). Recently, several variance reduction methods have been proposed and shown to be effective in practice, such as Q($\lambda$) (Harutyunyan et al., 2016), Retrace ($\lambda$) (Munos et al., 2016) and C-trace (Rowland et al., 2020), and so on (Espeholt et al., 2018; Horgan et al., 2018; Asis et al., 2017). In contrast to PI, Value Iteration (VI) methods propagate the values of the most promising actions backward one step at a time (Sutton & Barto, 2018; Szepesvári, 2010). Such methods can safely use data from any behavioral policy. However, step-by-step value propagation makes them somewhat ill-suited for general multi-step CA.

Here we provide a new tool for multi-step off-policy learning by extending VI approaches to the multi-step setting. The foundation of our method is a new Bellman operator, the *Highway Operator*, which connects current and future states through multiple "highways," and focuses on the most promising one. Highways are constructed through various policies looking ahead for multiple steps. Our operator has the following desirable properties: 1) It yields a new Bellman Optimality Equation that reflects the latent structure of multi-step CA, providing a novel sufficient condition for the optimal VF; 2) It effectively assigns future credit to past states across multiple time steps and has

remarkable convergence properties; 3) It yields a family of novel multi-step off-policy algorithms that do not need importance sampling, safely using arbitrary off-policy data. Experiments on toy tasks and visual MinAtar Games (Young & Tian, 2019) illustrate that our *Highway RL* algorithms outperform existing multi-step methods.

## 2  PRELIMINARIES

A *Markov Decision Processes (MDP)* (Puterman, 2014) is described by the tuple $\mathcal{M} = (\mathcal{S}, \mathcal{A}, \gamma, \mathcal{T}, \mu_0, \mathcal{R})$, where $\mathcal{S}$ is the state space; $\mathcal{A}$ is the action space; $\gamma \in [0, 1)$ is the discount factor. We assume MDPs with countable $\mathcal{S}$ (discrete topology) and finite $\mathcal{A}$. $\mathcal{T} : \mathcal{S} \times \mathcal{A} \to \Delta(\mathcal{S})$ is the transition probability function; $\mu_0$ denotes the initial state distribution; $\mathcal{R} : \mathcal{S} \times \mathcal{A} \to \Delta(\mathbb{R})$ denotes reward probability function. We use the following symbols to denote related conditional probabilities: $\mathcal{T}(s'|s, a)$, $\mathcal{R}(\cdot|s, a)$, $s, s' \in \mathcal{S}$, $a \in \mathcal{A}$. We also use $r(s, a) \triangleq \mathbb{E}_{R \sim \mathcal{R}(\cdot|s,a)}[R]$ for convenience. In order to make the space of value functions complete (assumption of Banach fixed point theorem), we assume bounded rewards, which with discounting produce bounded value functions. We denote $l_\infty(\mathcal{X})$ the space of bounded sequences with supremum norm $\|\cdot\|_\infty$ with support $\mathcal{X}$ assuming $\mathcal{X}$ is countable and has discrete topology. Completeness of our value spaces then follows from completeness of $l_\infty(\mathbb{N})$ [1].

The goal is to find a policy $\pi : \mathcal{S} \to \Delta(\mathcal{A})$ that yield maximal return. The return is defined as the accumulated discounted reward from time step $t$, i.e., $G_t = \sum_{n=0}^\infty \gamma^n r(s_{t+n}, a_{t+n})$. The state-value function (VF) of a policy $\pi$ is defined as the expected return of being in state s and following policy $\pi$, $V^\pi(s) \triangleq \mathbb{E}[G_t|s_t = s; \pi]$. Let $\Pi$ denote the space of all policies. The *optimal VF* is $V^* = \max_{\pi \in \Pi} V^\pi$. It is also convenient to define the action-VF, $Q^\pi(s, a) \triangleq \mathbb{E}[G_t|s_t = s, a_t = a; \pi]$ and the optimal action-VF is denoted as $Q^* = \max_{\pi \in \Pi} Q^\pi$. The Bellman Expectation/Optimality Equation and the corresponding operators are as follows:

$$\mathcal{B}^\pi V^\pi = V^\pi, \text{ where } (\mathcal{B}^\pi V)(s) \triangleq \mathbb{E}_{a \sim \pi(\cdot|s), s' \sim \mathcal{T}(\cdot|s,a)}\left[r(s, a) + \gamma V(s')\right] \tag{1}$$

$$\mathcal{B} V^* = V^*, \text{ where } (\mathcal{B} V)(s) \triangleq \max_a \left[r(s, a) + \gamma \mathbb{E}_{s' \sim \mathcal{T}(\cdot|s,a)}[V(s')]\right]. \tag{2}$$

## 3  HIGHWAY REINFORCEMENT LEARNING

Value Iteration (VI) looks ahead for one step to identify a promising value using only short-term information (see eq. 2). How can we quickly exploit long-term information through larger lookaheads? Our idea is to exploit the information conveyed by policies. We connect current and future states through multiple "highways" induced by policies, allowing for the unimpeded flow of credit across various lookaheads, and then focus on the most promising highway. Formally, we propose the novel *Highway Bellman Optimality Operator* $\mathcal{G}_\mathcal{N}^{\widehat{\Pi}}$ (Highway Operator in short), defined by

$$\mathcal{G}_\mathcal{N}^{\widehat{\Pi}} V(s_0) \triangleq \max_{\pi \in \widehat{\Pi}} \max_{n \in \mathcal{N}} \mathbb{E}_{\tau_{s_0}^n \sim \pi}\left[\sum_{t=0}^{n-1} \gamma^t r(s_t, a_t) + \gamma^n \max_{a_n'}\left[r(s_n, a_n') + \gamma \mathbb{E}_{s_{n+1}'}\left[V(s_{n+1}')\right]\right]\right], \tag{3}$$

where $\widehat{\Pi} = \{\pi_1, \cdots, \pi_m \cdots, \pi_M | \pi_m \in \Pi\}$ is a *set of behavioral policies*, which are used to collect data; $n$ is called the *lookahead depth* (also named *bootstrapping step* in RL literature), and $\mathcal{N}$ is the *set of lookahead depths*, which we assume always includes 0 ($0 \in \mathcal{N}$) unless explicitly stated otherwise; $\tau_{s_0}^n = (s_0, a_0, s_1, a_1, s_2, a_2, \cdots, s_n)$, and $\tau_{s_0}^n \sim \pi$ is the trajectory starting from $s_0$ by executing policy $\pi$ for $n$ steps. Fig. 1 (Left) illustrates the backup diagram of this operator. Our operator can be rewritten using the Bellman Operators:

$$\mathcal{G}_\mathcal{N}^{\widehat{\Pi}} V \triangleq \max_{\pi \in \widehat{\Pi}} \max_{n \in \mathcal{N}} (\mathcal{B}^\pi)^n \mathcal{B} V. \tag{4}$$

As implied by eq. (3) and eq. (4), given some trial, we pick a policy and a possible lookahead (up to the trial end if $N$ is sufficiently large) that maximize the cumulative reward during the lookahead

---

[1] The space of all bounded sequences with supremum norm, which is known to be a complete metric space.

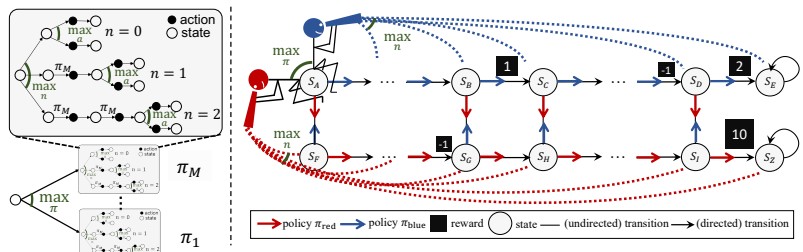

Figure 1: (Left) Backup diagram of Highway Operator $\mathcal{G}_{\mathcal{N}}^{\widehat{\Pi}}$ with $\mathcal{N} = \{0, 1, 2\}$. (Right) An illustrative example of "highway" under a simple $N$-horizon MDP, where $N$ is the horizon between the start state $s_A$ and the end state $s_Z$. With the highway (dashed lines) induced by rolling-out behavioral policies, the start state $s_A$ can directly access various deeper horizon information. By taking maximization over this information, the high credit information can be directly assigned to the previous states through the highway.

interval plus a dynamic programming-based estimate of the future return. This provides a highway to quickly transport credit information from the future into the past. That's why our operator generally converges faster than the classical Bellman Optimality Operator. See the detailed theoretical analysis in Section 5. Fig. 1 (Right) illustrates this highway through an example: an $N$-horizon MDP problem and two policies. A highway connects the start state $s_A$ to information derived from deeper lookaheads. In this example, our operator converges to the optimal VF within 2 iterations, while $\mathcal{B}$ needs $N$.

The following result shows that our Highway operator is a contraction on the complete metric space $l_\infty$ and admits $V^*$ as a unique fixed point.

**Theorem 1** *(Properties of Highway Operator) For any $\widehat{\Pi}$ and $\mathcal{N}$ (s.t. $0 \in \mathcal{N}$), we have*

*1) $\mathcal{G}_{\mathcal{N}}^{\widehat{\Pi}}$ is a contraction on complete metric space $l_\infty(\mathcal{S})$, i.e. for any $V, V' \in l_\infty(\mathcal{S})$, we have[2] $\|\mathcal{G}_{\mathcal{N}}^{\widehat{\Pi}} V - \mathcal{G}_{\mathcal{N}}^{\widehat{\Pi}} V'\| \leq \|\mathcal{B}V - \mathcal{B}V'\| \leq \gamma \|V - V'\|$;*

*2) (Highway Bellman Optimality Equation, Highway Equation in short) $V^*$ is the only fixed point $\mathcal{G}_{\mathcal{N}}^{\widehat{\Pi}}$, that is, for all $V \in l_\infty(S)$ holds $V = \mathcal{G}_{\mathcal{N}}^{\widehat{\Pi}} V$ if and only if $V = V^*$. Formally, we have*

$$\mathcal{G}_{\mathcal{N}}^{\widehat{\Pi}} V^* \triangleq \max_{\pi \in \widehat{\Pi}} \max_{n \in \mathcal{N}} (\mathcal{B}^\pi)^n \mathcal{B}V^* = V^*, \tag{5}$$

*3) For any $V_0 \in l_\infty(\mathcal{S})$ and any sequence of policy sets $(\widehat{\Pi}_k), k \in \mathbb{N}$, the sequence $(\mathcal{G}_{\mathcal{N}}^{\widehat{\Pi}_k} \circ \mathcal{G}_{\mathcal{N}}^{\widehat{\Pi}_{k-1}} \circ \ldots \circ \mathcal{G}_{\mathcal{N}}^{\widehat{\Pi}_1})[V_0], k \in \mathbb{N}$ converges R-linearly to $V^*$ with convergence rate $\gamma$.*

All the proofs are provided in Appendix A. Note that $0 \in \mathcal{N}$ is necessary for the guarantee of the fixed point property (Point 2), but not for the contraction property. This theorem implies that our Highway operator can provide a powerful extension to Bellman Optimality Equation, which can be potentially applied to various RL control methods. Table 1 summarizes the comparison to classical Bellman Operators and some advanced operators. More details on the comparison are in Section 5.

## 4 ALGORITHMS

Here we illustrate three applications of our Highway theory for model-based and model-free RL algorithms. Although we listed only three instances in this paper, note that our theory can be potentially applied to various RL methods which involve value estimation, such as actor-critic methods.

---

[2]We denote by $\| \cdot \|$ the supremum norm throughout the paper.

| Type | Operator | Fixed Point | Guaranteed convergence | Convergence Rate | NOT suffering variance explosion of IS ratios |
|---|---|---|---|---|---|
| Classical Operators | Bellman Optimality Operator $\mathcal{B}$ (eq. 2) | $V^*/Q^*$ | ✓ | $\gamma$ | ✓ |
| | Bellman Expectation Operator $\mathcal{B}^{\pi'}$ (eq. 1) | $V^{\pi'}/Q^{\pi'}$ | ✓ | $\gamma$ | ✓ |
| Multi-Step Off-Policy Operators | † Multi-Step Bellman Optimality Operator ($N \geq 2$) $\mathbb{E}_{\pi \sim \mathcal{P}_{\widehat{\Pi}}}\left[(\mathcal{B}^\pi)^{N-1}\mathcal{B}\right]$ (Hessel et al., 2018; Horgan et al., 2018) | $V^*/Q^*$ | For $\widehat{\Pi}$ s.t. $\forall \pi \in \widehat{\Pi}, \pi = \pi^*$ | $\gamma^N$ | ✓ |
| | § Multi-Step IS-based Bellman Expectation Operator $\mathbb{E}_{\pi \sim \mathcal{P}_{\widehat{\Pi}}}\left[(\breve{\mathcal{B}}_\pi^{\pi'})^N\right]$ (Sutton & Barto, 2018) | $V^{\pi'}/Q^{\pi'}$ | For any $\widehat{\Pi}, N$ | $\gamma^N$ | ✗ |
| | Q($\lambda$) Operator (Harutyunyan et al., 2016) | $V^{\pi'}/Q^{\pi'}$ | For $\widehat{\Pi}$ s.t. $\forall \pi \in \widehat{\Pi}, \pi$ is close to $\pi'$ | $\gamma$ | ✓ |
| | Retrace($\lambda$) Operator (Munos et al., 2016) | $V^{\pi'}/Q^{\pi'}$ | For any $\widehat{\Pi}$ | $\gamma$ | ✓ |
| Highway Operator and its Variants (Ours) | **Highway Operator** (eq. 4) $\mathcal{G}_{\mathcal{N}}^{\widehat{\Pi}} \triangleq \max_{\pi \in \widehat{\Pi}} \max_{n \in \mathcal{N}} (\mathcal{B}^\pi)^n \mathcal{B}$ | $V^*/Q^*$ | For any $\widehat{\Pi}, \mathcal{N}$ | $\gamma$ ($\gamma^2, \gamma^N$ under some conditions) | ✓ |
| | **Softmax Highway Operator** (eq. 9) $\widetilde{\mathcal{G}}_{\mathcal{N}}^{\widehat{\Pi}} \triangleq smax^\alpha_{\pi \in \widehat{\Pi}} smax^\alpha_{n' \in \mathcal{N}} \max_{n \in \{0,n'\}} (\mathcal{B}^\pi)^n \mathcal{B}$ | $V^*/Q^*$ | For any $\widehat{\Pi}, \mathcal{N}, \alpha$ | $\gamma$ | ✓ |
| | ‡ **Expectation Highway Operator** $\overline{\mathcal{G}}_{\widehat{\Pi}}^{\mathcal{N}} \triangleq \mathbb{E}_{\pi \sim \mathcal{P}_{\widehat{\Pi}}}\left[\max_{n \in \mathcal{N}} (\mathcal{B}^\pi)^n \mathcal{B}\right]$ | $V^*/Q^*$ | For any $\widehat{\Pi}, \mathcal{N}$ | $\gamma$ | ✓ |

Table 1: Properties of the operators. $\widehat{\Pi}$ denote the set of behavioral policies; $\mathcal{P}_{\widehat{\Pi}}$ is a distribution over $\widehat{\Pi}$ and $\mathcal{P}_{\widehat{\Pi}}(\pi)$ is the probability of selecting $\pi$; $\pi'$ denotes the target policy of policy evaluation. † and ‡: Please refer to Appendix A.2 for details. §: $\breve{\mathcal{B}}_\pi^{\pi'}$ is the importance sampling-based (IS-based) Bellman Expectation Operator (see Appendix A.4).

## 4.1 Model-based Reinforcement Learning

**Highway Value Iteration.** From the new operator, we can naturally derive a new Value Iteration algorithm, (Algorithm B.1). Specifically, for a finite $\mathcal{S}$, the update using the Highway Operator $\mathcal{G}_{\mathcal{N}}^{\widehat{\Pi}}$ (eq. 3) can be written as

$$\mathbf{v}_{k+1} = \max_{\pi \in \widehat{\Pi}} \max_{n \in \mathcal{N}} \overbrace{\left[\underbrace{\sum_{i=0}^{n}\left[(\gamma\mathbf{T}^\pi)^{i-1}\right]\mathbf{r}^\pi}_{\textbf{PART}1} + \underbrace{(\gamma\mathbf{T}^\pi)^n}_{\textbf{PART}2}\left[\max_a\left[\mathbf{r}^a + \gamma\mathbf{T}^a\mathbf{v}_k\right]\right]\right]}^{\textbf{PART}3}, \quad (6)$$

where $\mathbf{v}_k$ is a $|\mathcal{S}| \times 1$ column vector of VF; $\mathbf{r}^a$ and $\mathbf{r}^\pi$ are $|\mathcal{S}| \times 1$ column vectors of rewards for action $a$ and policy $\pi$ respectively, where $[\mathbf{r}^a]_s = r(s,a)$, $[\mathbf{r}^\pi]_s = \sum_a \pi(a|s)r(s,a)$. $\mathbf{T}^a$ and $\mathbf{T}^\pi$ are $|\mathcal{S}| \times |\mathcal{S}|$ matrices of transition probabilities for action $a$ and policy $\pi$ respectively, where $[\mathbf{T}^a]_{s,s'} = \mathcal{T}(s'|s,a)$, $[\mathbf{T}^\pi]_{s,s'} = \sum_a \pi(a|s)\mathcal{T}(s'|s,a)$. The computational complexity of each iteration is $\mathcal{O}\left(\left(|\mathcal{A}| + \left|\widehat{\Pi}\right||\mathcal{N}|\right)|\mathcal{S}|^2\right)$. Two strategies can be adopted to accelerate the update process. First, the matrix in **PART 1** and **PART 2** in eq. (6) can be computed and reused for each $\pi$ and $n$, as they are fixed during the iteration process. Second, **PART 3** can be computed in parallel for each policy $\pi \in \widehat{\Pi}$ and $n \in \mathcal{N}$.

## 4.2 Off-Policy Learning in model-free Reinforcement Learning

In model-free RL, it is convenient to use $Q$ instead of $V$. The corresponding operator[3] is defined as

---

[3]We use the same notation $\mathcal{G}_{\mathcal{N}}^{\widehat{\Pi}}$ to denote the operator w.r.t. the the VF $V$ and the action VF $Q$ when there is no ambiguity. Similarly, for the Bellman operators, we will reuse the same symbols.

$$\mathcal{G}_{\mathcal{N}}^{\widehat{\Pi}} Q(s_0, a_0) \triangleq \max_{\pi \in \widehat{\Pi}} \max_{n \in \mathcal{N}} \mathbb{E}_{\tau_{s_0,a_0}^{n+1} \sim \pi} \underbrace{\left[ \sum_{t=0}^{n} \gamma^t r_t + \gamma^{n+1} \max_{a'_{n+1}} Q\left( s_{n+1}, a'_{n+1} \right) \right]}_{G_Q^{n+1}(\tau_{s_0,a_0}^{n+1})}, \tag{7}$$

where $G_Q^{n+1}(\tau_{s_0,a_0}^{n+1})$ is the $n+1$-step return; $\tau_{s_0,a_0}^{n+1} \triangleq (s_0, a_0, r_0, s_1, a_1, r_1, \cdots, s_{n+1})$. This operator can also be represented by the Bellman Operators, i.e., $\mathcal{G}_{\mathcal{N}}^{\widehat{\Pi}} Q \triangleq \max_\pi \max_n (\mathcal{B}^\pi)^n \mathcal{B} Q$; and it converges to the optimal action VF $Q^*$ with any set of behavioral policies $\widehat{\Pi}$, i.e., $\mathcal{G}_{\mathcal{N}}^{\widehat{\Pi}} Q^* = Q^*$ (similar to Theorem 1, see APPENDIX Theorem 6 for a formal statement). This means that it can utilize any off-policy data collected by arbitrary policy, without additional corrections. We propose two methods for the tabular VF and VF approximation, named Highway Q-Learning and Highway DQN respectively.

**Highway Q-Learning.** Let $\mathcal{D}_{s_0,a_0}^{(m)} = \{\tau_{s_0,a_0}^{n+1} | \tau_{s_0,a_0}^{n+1} \sim \pi_m\}$ denote the trajectory data collected by the policy $\pi_m$. The $k$-th VF $Q_k$ is updated in the following way:

$$Q_{k+1}(s_0, a_0) = \max_{m \in \mathsf{M}_{s_0,a_0}} \max_{n \in \mathcal{N}} \widehat{\mathbb{E}}^{\mathcal{D}_{s_0,a_0}^{(m)}} [G_{Q_k}^{n+1}(\tau_{s_0,a_0}^{n+1})], \tag{8}$$

where $\mathsf{M}_{s_0,a_0} \subseteq \left\{ m | |\mathcal{D}_{s_0,a_0}^{(m)}| \neq 0 \right\}$ is a subset of indexes of the dataset that are not empty under $(s_0, a_0)$; $\widehat{\mathbb{E}}^{\mathcal{D}_{s_0,a_0}^{(m)}}[\cdot] = \frac{1}{\left|\mathcal{D}_{s_0,a_0}^{(m)}\right|} \sum_{\tau_{s_0,a_0}^{n+1} \in \mathcal{D}_{s_0,a_0}^{(m)}} [\cdot]$ is the empirical averaged value. Note that all we need to do is saving the trajectory data into the corresponding datasets $\mathcal{D}_{s_0,a_0}^{(m)}$, and then search over them, without having to know the form of $\pi_m$ or save the $\pi_m$ into the set of behavioral policies $\widehat{\Pi}$. The algorithm, *Highway Q-Learning*, is presented in Appendix B Algorithm B.3.

**Highway DQN.** For large-scale space or continuous MDPs, the VF is usually approximated. Due to the estimation noise involved in function approximation (Van Hasselt et al., 2016), our method may lead to overestimation by the two maximization operations (over policies and lookahead depths). However, as we show below, this issue can be easily solved through a minor modification.

We propose a new variant of eq. (7), named *Softmax Highway Operator*, as follows

$$\widetilde{\mathcal{G}}_{\mathcal{N}}^{\widehat{\Pi}} Q(s_0, a_0) \triangleq smax^\alpha_{\pi \in \widehat{\Pi}} smax^\alpha_{n' \in \mathcal{N}} \max_{n \in \{0, n'\}} \mathbb{E}_{\tau_{s_0,a_0}^{n+1} \sim \pi} \left[ G_Q^{n+1}(\tau_{s_0,a_0}^{n+1}) \right], \tag{9}$$

where the softmax operator $smax^\alpha$ with the temperature parameter $\alpha$ is defined as

$$smax^\alpha_{x \in \mathcal{X}} f(x) \triangleq \sum_{x \in \mathcal{X}} \frac{\exp(\alpha f(x))}{\sum_{x' \in \mathcal{X}} \exp(\alpha f(x'))} f(x), \tag{10}$$

$smax^\alpha$ reduces to $\max$ when $\alpha \to \infty$. We have the following theorem.

**Theorem 2** *For any $\alpha$, any $\widehat{\Pi}$, and any $\mathcal{N}$, we have $\widetilde{\mathcal{G}}_{\mathcal{N}}^{\widehat{\Pi}} Q^* = Q^*$ and $(\forall Q \in l_\infty(\mathcal{S} \times \mathcal{A}))$ :* $\|\widetilde{\mathcal{G}}_{\mathcal{N}}^{\widehat{\Pi}} Q - Q^*\| \leq \gamma \|Q - Q^*\|$.

The operator in eq. (9) is derived in the following way. First, the Highway $Q$ Operator $\mathcal{G}_{\mathcal{N}}^{\widehat{\Pi}}$ in eq. (7) can be rewritten as an equivalent form, $\mathcal{G}_{\mathcal{N}}^{\widehat{\Pi}} Q \triangleq \max_{\pi \in \widehat{\Pi}} \max_{n' \in \mathcal{N}} \max_{n \in \{0, n'\}} (\mathcal{B}^\pi)^n \mathcal{B} Q$. Then, we replace the first two max operators $\max_\pi \max_{n'}$ with the softmax operators $smax^\alpha_\pi smax^\alpha_{n'}$.

The above modification of Softmax Highway Operator $\widetilde{\mathcal{G}}_{\mathcal{N}}^{\widehat{\Pi}}$ is necessary to a) remain unbiased w.r.t. $Q^*$ (as shown in APPENDIX Theorem 2)[4] ; b) alleviate the overestimation issue and improve exploration with the softmax operation, which has been shown effective in recent RL literature (Fox et al., 2015; Haarnoja et al., 2017; Schulman et al., 2017; Song et al., 2019).

---

[4]Note another variant $smax^\alpha_\pi smax^\alpha_n (\mathcal{B}^\pi)^n \mathcal{B} Q$ (without $\max_{n \in \{0, n'\}}$) is generally biased w.r.t. $Q^*$. Please refer to Appendix A.3 for the detail of the reason.

Based on the above theoretically justified operator, we propose the following objective function for updating the action VF $Q_\theta(s, a)$ parametrized by the parameter $\theta$:

$$L\left(\theta\right) = \sum_{(s_0,a_0)\in\widehat{\mathcal{D}}} \left[ Q_\theta\left(s_0, a_0\right) - \operatorname*{smax}_{m\in\mathsf{M}_{s_0,a_0}}^{\alpha} \operatorname*{smax}_{n'\in\mathcal{N}}^{\alpha} \max_{n\in\{0,n'\}} \widehat{\mathbb{E}}^{\mathcal{D}_{s_0,a_0}^{(m)}} \left[ G_{Q_{\theta'}}^{n+1}(\tau_{s_0,a_0}^{n+1}) \right] \right]^2, \tag{11}$$

where $Q_{\theta'}$ is the target network parametrized by $\theta'$, occasionally copied from $\theta$; $\widehat{\mathcal{D}} = \{(s_0, a_0)\})$ is the sampled batch data of state-action $(s_0, a_0)$ pairs. The computational complexity of the method implied by the equation above is close to the one of existing eligibility trace-based methods (Schulman et al., 2016; Munos et al., 2016), which also need to compute $n$-step returns $G_Q^n$ for each $n$. The resulting algorithm is called *Highway DQN*, presented in APPENDIX Algorithm B.4.

In practice, our algorithm balances the trade-off between accuracy and sample efficiency by deciding the number of trials per policy, the size of the search space (of behavioral policies and lookahead depths), and the softmax temperature. While more trials per policy may improve the estimation accuracy, they may also cost more samples and reduce sample efficiency. On the other hand, while a larger search space may increase efficiency, it might incur overestimation issues when the estimate is biased, leading to high variance.

In summary, our Highway Q-Learning and Highway DQN can recycle any trajectory data collected by some arbitrary policy, and utilize the multiple-step trajectory data that do not require Importance Sampling-based corrections (as stated in Theorem 1 and 2).

## 5 THEORETICAL ANALYSIS

In this section, we study the theoretical properties of Highway operator[5] $\mathcal{G}_\mathcal{N}^{\widehat{\Pi}}$ and show its superiority over classical Bellman operators, e.g., $\mathcal{B}$ and $\mathcal{B}^\pi$. For convenience, our analysis is in the space of state VF, *i.e.*, the operators $\mathcal{G}_\mathcal{N}^{\widehat{\Pi}}, \mathcal{B}, \mathcal{B}^\pi, \ldots$ are assumed to be mappings on $l_\infty(\mathcal{S})$.

First, we compare our Highway Operator $\mathcal{G}_\mathcal{N}^{\widehat{\Pi}}$ to the Bellman Optimality Operator $\mathcal{B}$.

**Theorem 3** *(Comparison to Bellman Optimality Operator $\mathcal{B}$) For all $V, V_0 \in l_\infty(\mathcal{S})$ holds:*
*1)* $\left\| \mathcal{G}_\mathcal{N}^{\widehat{\Pi}} V - V^* \right\| \leq \|\mathcal{B}V - V^*\|$;
*2) Assume $V_0 \leq V^*$. For any $s$ we have $\left| \mathcal{G}_\mathcal{N}^{\widehat{\Pi}} V_0\left(s\right) - V^*\left(s\right) \right| \leq |\mathcal{B}V_0\left(s\right) - V^*\left(s\right)|$, where the strict inequality holds as long as there exists $\pi' \in \widehat{\Pi}$ such that $\arg\max_{n\in\mathcal{N}} \left( \mathcal{B}^{\pi'} \right)^n \mathcal{B}V_0\left(s\right) > 0$.*

The first point of the theorem implies that our operator converges at the same rate as the Bellman Optimality Operator in the worst case. The second point shows a state-wise convergence comparison under the case of $V_0 \leq V^*$. Our operator generally converges faster than the Bellman Optimality Operator as long as one behavioral policy finds a better path by looking forward for $n$ steps ($n > 0$). Note that the condition $V_0 \leq V^*$ can be easily satisfied by setting $V_0 = \min_{s',a'} r(s', a')$. Moreover, as long as $V_0 \leq V^*$, we have $(\mathcal{G}_\mathcal{N}^{\widehat{\Pi}})^{\circ k} V_0 \leq V^*$ for any $k$ (see Lemma 2 in Appendix).

Then, we show the relation of our Highway operator with the Multi-step Bellman Expectation operator $\mathcal{B}_N^\pi \triangleq (\mathcal{B}^\pi)^{\circ N}$, which is adopted in Generalized Policy Iteration (GPI) (see APPENDIX Algorithm B.2). GPI needs to balance the evaluation-improvement trade-off by adapting hyperparameter $N$ in $(\mathcal{B}^\pi)^{\circ N}$. Our Highway operator provides an optimal solution for deciding such hyperparameter $N$ in terms of approaching the optimal VF.

**Theorem 4** *(Comparison to Multi-Step Bellman Expectation Operator) Assume that the VF $V_k \leq V^*$. Let $V_{k+1}^{\mathcal{G}_\mathcal{N}^{\widehat{\Pi}_k}}$ and $V_{k+1}^{\mathcal{B}_N^{\pi_k}}$ denote the $k + 1$-th VF of Highway Value Iteration with hyperparameter*

---

[5]Note that, unless otherwise stated, the results hold for any set of behavioral policies $\widehat{\Pi}$ and set of lookahead depths $\mathcal{N}$. For convenience, we analyze under fixed $\widehat{\Pi}$. However, these results can also be extended to the case of dynamically changing $\widehat{\Pi}$ (as shown Highway Value Iteration in Algorithm B.1, where $\widehat{\Pi}_k$ could change over different $k$-th iterations as new policies are added to the set of behavioral policies).

$\mathcal{N}$ and Generalized Policy Iteration with hyperparameter $N$. We have

$$\left\| V_{k+1}^{\mathcal{G}_{\mathcal{N}}^{\widehat{\Pi}_k}} - V^* \right\| \leq \min_{N \in \mathcal{N}} \left\| V_{k+1}^{\mathcal{B}_{N+1}^{\pi_k}} - V^* \right\| \tag{12}$$

Next, we show that by assuming that some of the behavioral policies act optimally within a few time steps, our Highway operator can achieve better convergence rates.

**Assumption A**$(\widehat{\Pi}, n)$  *Given set of behavioral policies $\widehat{\Pi}$, a lookahead depth $n$. Let $\mathcal{S}_{s,n}^{\pi'}$ denote the set of all possible visited states by executing policy $\pi'$ for $n$ steps from state $s$. For each $s \in \mathcal{S}$, there exists at least one policy $\pi' \in \widehat{\Pi}$ such that $\pi'(s') = \pi^*(s')$ for any $s' \in \mathcal{S}_{s,n}^{\pi'}$ where $\pi^*$ refers to an optimal policy. Note the quantification order: $(\forall s \in \mathcal{S}, \exists \pi' \in \widehat{\Pi}, \forall s' \in \mathcal{S}_{s,n}^{\pi'}) : \pi'(s') = \pi^*(s')$.*

**Theorem 5**  *(Better Contraction Rate) Assume $N-1 \in \mathcal{N}$, $\widehat{\Pi}$ satisfies Assumption $A(\widehat{\Pi}, N-1)$, and $V_0 \in l_\infty(\mathcal{S})$ and $V_0 \leq V^*$, the convergence rate of $\mathcal{G}_{\mathcal{N}}^{\widehat{\Pi}}$ is $\gamma^N$, i.e., $\|\mathcal{G}_{\mathcal{N}}^{\widehat{\Pi}} V - V^*\| \leq \gamma^N \|V - V^*\|$;*

This theorem implies that when the set of behavioral policies $\widehat{\Pi}$ satisfies Assumption A$(\widehat{\Pi}, N-1)$, our operator $\mathcal{G}_{\mathcal{N}}^{\widehat{\Pi}}$ has a convergence rate of $\gamma^N$. Note that, in Assumption A$(\widehat{\Pi}, N-1)$, $\widehat{\Pi}$ is not required to include an optimal policy $\pi^*$. Instead, it only requires that, for each state, there exists one policy that behaves well within a period ($N-1$ consecutive steps), and the well-behaved policy could be varying towards different starting from each state. Note that this condition is much weaker than having some optimal or near-optimal policies included in the behavioral policy set. Specifically, Assumption A$(\widehat{\Pi}', 1)$ can be satisfied by constructing a $\widehat{\Pi}' = \{\pi_a | a \in \mathcal{A}, (\forall s \in \mathcal{S}) : \pi_a(a|s) = 1\}$, yielding a convergence rate of $\gamma^2$.

Note that although the operator $(\mathcal{B})^{\circ N}$ also leads to a convergence rate of $\gamma^N$, our Highway operator $\mathcal{G}_{\mathcal{N}}^{\widehat{\Pi}}$ differs essentially from it. A major difference is that Highway operator $\mathcal{G}_{\mathcal{N}}^{\widehat{\Pi}}$ applies Multi-step Bellman Expectation Operator $(\mathcal{B}^\pi)^{\circ n}$ instead of applying Multi-step Bellman Optimality Operator $(\mathcal{B})^{\circ N}$. In the model-free case, $(\mathcal{B}^\pi)^{\circ n} V$ can utilize the $n$-step trajectory data generated by $\pi$ with minor cost (just by accumulating rewards within $n$-step, i.e., $\sum_{n'} \gamma^{n'} r_{t+n'} + \gamma^n V(s_{t+n})$). While $(\mathcal{B})^{\circ N}$ can only utilize the 1-step data and needs to update the VF $N$ times (implying by eq. 2).

# 6  RELATED WORK

Multi-step RL methods has been studied in RL for a long history, including multi-step SARSA (Sutton & Barto, 2018), Tree Backup (Precup, 2000), Q($\sigma$) (Asis et al., 2017), and Monte Carlo methods (which can be regarded as $\infty$-step learning). $\lambda$-return assign exponentially decaying weights depends on the decay factor $\lambda$ (Sutton & Barto, 2018; Schulman et al., 2016; White & White, 2016). Sahil et al. (2017) proposed a more general form called weighted returns, which assigns weights to all $n$-step returns. Roughly, the lookahead depth, the decay factor, and the weights represent the prior knowledge or bias regarding appropriate CA intervals, usually tuned in a case-by-case manner. Our method instead adaptively adjusts the lookahead depth in line with the quality of the data and the learned VF. Another similar work related to CA is RUDDER (Arjona-Medina et al., 2019), which trains an LSTM to re-assign credits to previous actions. Our methods derive a simple but sound principle for transporting the credit with minor cost.

Existing off-policy learning methods usually use additional corrections for off-policy data. Classical importance sampling (IS)-based methods suffer from high variance due to the products of IS ratios (Cortes et al., 2010; Metelli et al., 2018). Several variance reduction techniques have been proposed. Munos et al. (2016)'s Retrace($\lambda$) reduces the variance by clipping the IS ratios, and has achieved great success in practice. Other work ignores IS: TB($\lambda$) (Precup, 2000) corrects the off-policy data by computing expectations w.r.t. data and estimated VF, using the probabilities of the target policy. Harutyunyan et al. (2016)'s Q($\lambda$) corrects the off-policy data by adding an off-policy correction term. Our method provides an alternative IS-free tool for off-policy learning. Similar to Retrace($\lambda$), it can safely use arbitrary off-policy data. Moreover, it is very efficient, offering a faster convergence rate under mild conditions. More research is needed, however, to understand how the variance of our method compares to that of advanced IS-based variance reduction methods like Retrace($\lambda$).

(a) Multi-Room Environment      (b) Choice    (c) Trace Back

Figure 2: (a) shows results of model-based algorithms in Multi-Room environments. The x-axis is the number of rooms. The y-axes for the three figures are total iteration, total samples, and total running time required by the algorithm, respectively. (b) and (c) in Choice and Trace Back environments: number of episodes required to solve the task, as a function of the delay of reward. Average over 100 seeds, 1 standard deviation.

Searching over various policies and (or) lookahead depths to improve the convergence of RL systems is an active field. Barreto et al. (2020) search over various policies while using only a fixed lookahead depth until the end of the trial. He et al. (2017) first propose to search over various lookahead depths along the trajectory data, but they do not search over various policies. Moreover, they employ the lookahead returns to construct additional inequality bounds on the VF. Our work instead contributes to a novel Bellman operator on updating the VF directly and provides a thorough theoretical analysis of the convergence properties. We also address that greedy operations (Barreto et al., 2020; He et al., 2017) may cause overestimation, by proposing a novel softmax operation to alleviate this issue in an unbiased fashion. Compared to previous methods searching over the product of original policy space and action space (Efroni et al., 2018; 2019; 2020; Jiang et al., 2018; Tomar et al., 2020), ours has a smaller search space (i.e., a limited set of policies). In summary, this paper contributes to a theoretically-justified Bellman operator which searches both policies and lookahead depths, leading to more flexibility and scalability against different settings and applications.

Our method can also be viewed as combining the best of both worlds: (1) direct policy search based on policy gradients (Williams, 1992) or evolutionary computation (Rechenberg, 1971) (where lookahead equals trial length—no runtime-based identification of useful subprograms and subgoals), and (2) dynamic programming-based (Bellman, 1957) identification of useful subgoals during runtime. This naturally and safely combines best subgoals/sub-policies (Schmidhuber, 1991) derived from data collected by arbitrary previously encountered policies. It can be viewed as a soft hierarchical chunking method (Schmidhuber, 1992) for rapid CA, with a sound Bellman-based foundation.

## 7 EXPERIMENTS

We designed our experiments to evaluate the algorithms under different cases and investigate their properties. Please refer to Appendix C for additional details of the experiment settings.

**Model-based Toy Tasks.** We first evaluate on a model-based task: Multi-Room environments with different number of rooms. The agent needs to go through many rooms and reach the goal to get a reward. We compare our Highway Value Iteration to classical Value Iteration (VI) and Policy Iteration (PI). The algorithms are evaluated until convergence. As shown in the three plots of Fig. 2 (a), our Highway Value Iteration outperforms VI and PI in terms of number of iterations required, total number of samples (total number of queries to the MDP model), and computation time.

**Model-free Toy Tasks.** To evaluate credit assignment efficiency, we evaluate the algorithms on two toy tasks involving delayed rewards (Arjona-Medina et al., 2019), in which a reward is only provided at the end of each trial. We compare our method to classical eligibility trace methods including $Q(\lambda)$, $SARSA(\lambda)$, Monte Carlo methods; and also the advanced credit assignment method RUDDER.

As shown in Fig. 2 (b) and (c), our method significantly outperforms all competitors on both tasks. For example, on Trace Back, our method requires only 20 episodes to solve the task, while the second best algorithm RUDDER requires more than 1000. Notably, the costs of Highway Q-Learning do not observably increase with the reward delays. In contrast, other methods such as $Q(\lambda)$ require exponentially increasing numbers of trials.

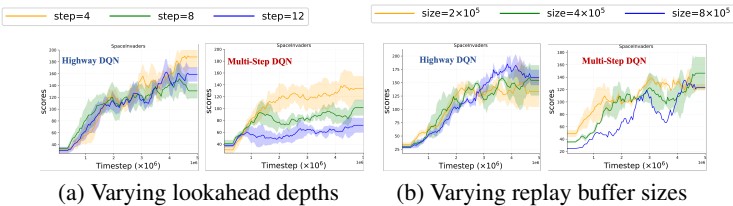

Figure 3: Episode rewards during the training on MinAtar Games (Young & Tian, 2019). Average over 5 seeds, 1 standard deviation.

(a) Varying lookahead depths      (b) Varying replay buffer sizes

Figure 4: The performance of Highway DQN and Multi-Step DQN using (a) varying lookahead depth and (b) varying replay buffer sizes. Average over 5 seeds, 1 standard deviation.

**MinAtar Games.** We evaluate our algorithms on benchmark tasks from MinAtar (Young & Tian, 2019). We compare to several advanced multi-step off-policy methods, including *Multi-Step DQN* (Horgan et al., 2018; Barth-Maron et al., 2018) and *Retrace(λ)* Munos et al. (2016). All multi-step methods are implemented on top of *Maxmin DQN* (Lan et al., 2020), which chooses the smallest Q-value among multiple target Q networks. All the compared methods adopt the same implementations to ensure that we measure differences between algorithms, not implementations.

Fig. 3 shows the performance of the algorithms. Highway DQN significantly outperforms all competitors in terms of both reward and sample efficiency on almost all the tasks. Compared to the advanced Retrace(λ), our method performs significantly better on 3 of 5 tasks while performing on par with it on the remaining 2 tasks.

**Ablation Study.** We conducted the following ablation studies to investigate properties of Highway DQN. (1) Lookahead depth ($\mathcal{N} = \{0, 1, \cdots, N-1\}$ for Highway DQN and $N$ for Multi-step DQN). As shown in Fig. 4a, compared to Multi-step DQN, our Highway DQN shows strong robustness against variations of the lookahead depth. This is because Highway DQN can adaptively choose the lookahead depth. (2) Replay buffer size. As shown in Fig. 4b, when the memory size increases to $8 \times 10^5$ (orange line), our Highway DQN shows a performance improvement, while Multi-Step DQN shows a degradation. (3) For the results with varying softmax Temperature $\alpha$ and number of target networks, please refer to APPENDIX Appendix C.4 for more details.

## 8 CONCLUSIONS

We introduced a novel multi-step Bellman Optimality Equation for efficient multi-step credit assignment (CA) in reinforcement learning (RL). We proved that its solution is the optimal value function and that the corresponding policy adjustments generally converge faster than the traditional Bellman Optimality operator. Our *Highway RL* methods combine the best of direct policy search (where CA is performed without trying to identify useful environmental states or subgoals during runtime), and standard RL, which finds useful states/subgoals through dynamic programming. Highway RL quickly and safely extracts useful sub-policies derived from data obtained through previously tested policies. The derived algorithms have several advantages over existing off-policy algorithms. Their feasibility and effectiveness were experimentally illustrated on a series of standard benchmark datasets. Future work will theoretically analyze the algorithm's behavior in the model-free case.

ETHICS STATEMENT

This paper provides a new operator for off-policy learning-based methods. The authors do not find any particular concerns w.r.t the potential ethical or societal consequence.

REPRODUCIBILITY STATEMENT

The code is available at `https://anonymous.4open.science/r/Highway-Reinforcement-Learning-4202` and more implementation details can be found at Appendix B and C.

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

## A    THEOREM PROOFS

### A.1    PROOF OF PROPERTIES OF HIGHWAY OPERATOR $\mathcal{G}_{\mathcal{N}}^{\widehat{\Pi}}$

**Theorem 1** *(Properties of Highway Operator) For any $\widehat{\Pi}$ and $\mathcal{N}$ (s.t. $0 \in \mathcal{N}$), we have*

*1) $\mathcal{G}_{\mathcal{N}}^{\widehat{\Pi}}$ is a contraction on complete metric space $l_{\infty}(\mathcal{S})$, i.e. for any $V, V' \in l_{\infty}(\mathcal{S})$, we have[6]*
$\|\mathcal{G}_{\mathcal{N}}^{\widehat{\Pi}}V - \mathcal{G}_{\mathcal{N}}^{\widehat{\Pi}}V'\| \leq \|\mathcal{B}V - \mathcal{B}V'\| \leq \gamma \|V - V'\|$;

*2) (Highway Bellman Optimality Equation, Highway Equation in short) $V^*$ is the only fixed point $\mathcal{G}_{\mathcal{N}}^{\widehat{\Pi}}$, that is, for all $V \in l_{\infty}(S)$ holds $V = \mathcal{G}_{\mathcal{N}}^{\widehat{\Pi}}V$ if and only if $V = V^*$. Formally, we have*

$$\mathcal{G}_{\mathcal{N}}^{\widehat{\Pi}}V^* \triangleq \max_{\pi \in \widehat{\Pi}} \max_{n \in \mathcal{N}} (\mathcal{B}^\pi)^n \mathcal{B}V^* = V^*, \tag{13}$$

*3) For any $V_0 \in l_{\infty}(\mathcal{S})$ and any sequence of policy sets $(\widehat{\Pi}_k), k \in \mathbb{N}$, the sequence $(\mathcal{G}_{\mathcal{N}}^{\widehat{\Pi}_k} \circ \mathcal{G}_{\mathcal{N}}^{\widehat{\Pi}_{k-1}} \circ \ldots \circ \mathcal{G}_{\mathcal{N}}^{\widehat{\Pi}_1})[V_0], k \in \mathbb{N}$ converges R-linearly to $V^*$ with convergence rate $\gamma$.*

*Proof:* 1) The contraction property can be obtained:

$$\left\| \mathcal{G}_{\mathcal{N}}^{\widehat{\Pi}}V - \mathcal{G}_{\mathcal{N}}^{\widehat{\Pi}}V' \right\| = \left\| \max_{\pi \in \widehat{\Pi}} \max_{n \in \mathcal{N}} (\mathcal{B}^\pi)^n \mathcal{B}V - \max_{\pi \in \widehat{\Pi}} \max_{n \in \mathcal{N}} (\mathcal{B}^\pi)^n \mathcal{B}V' \right\|$$
$$\leq \max_{\pi \in \widehat{\Pi}} \max_{n \in \mathcal{N}} \left\| (\mathcal{B}^\pi)^n \mathcal{B}V - (\mathcal{B}^\pi)^n \mathcal{B}V' \right\|$$
$$\leq \max_{\pi \in \widehat{\Pi}} \max_{n \in \mathcal{N}} \gamma^n \left\| \mathcal{B}V - \mathcal{B}V' \right\|$$
$$\leq \left\| \mathcal{B}V - \mathcal{B}V' \right\|$$
$$\leq \gamma \left\| V - V' \right\|$$

---

[6] We denote by $\| \cdot \|$ the supremum norm throughout the paper.

2) By application of the Banach fixed point theorem to point 1., the operator $\mathcal{G}_{\mathcal{N}}^{\widehat{\Pi}}$ must have a unique fixed point. Therefore, it suffices to verify that $V^*$ is a fixed point of $\mathcal{G}_{\mathcal{N}}^{\widehat{\Pi}}$. Our operator can be rewritten as

$$\mathcal{G}_{\mathcal{N}}^{\widehat{\Pi}} V \triangleq \max_{\pi \in \widehat{\Pi}} \max_{n \in \mathcal{N}} \left(\mathcal{B}^\pi\right)^n \mathcal{B} V.$$

Using the fact that $\mathcal{B}^\pi V^* \leq \mathcal{B} V^* = V^*$ and $\mathcal{B}^\pi$ monotony we obtain $\left(\mathcal{B}^\pi\right)^n \mathcal{B} V^* \leq V^*$ for any $\pi$ and $n \geq 0$ (with the equality when $n = 0$), which implies

$$\max_{\pi \in \widehat{\Pi}} \max_{n \in \mathcal{N}} \left(\mathcal{B}^\pi\right)^n \mathcal{B} V^* = V^*.$$

3) From 1) and 2) follows: $\|(\mathcal{G}_{\mathcal{N}}^{\widehat{\Pi}_k} \circ \mathcal{G}_{\mathcal{N}}^{\widehat{\Pi}_{k-1}} \circ \ldots \mathcal{G}_{\mathcal{N}}^{\widehat{\Pi}_1})[V_0] - V^*\| = \|(\mathcal{G}_{\mathcal{N}}^{\widehat{\Pi}_k} \circ \mathcal{G}_{\mathcal{N}}^{\widehat{\Pi}_{k-1}} \circ \ldots \mathcal{G}_{\mathcal{N}}^{\widehat{\Pi}_1})[V_0] - \mathcal{G}_{\mathcal{N}}^{\widehat{\Pi}_k} V^*\| \leq \gamma \|(\mathcal{G}_{\mathcal{N}}^{\widehat{\Pi}_{k-1}} \circ \ldots \mathcal{G}_{\mathcal{N}}^{\widehat{\Pi}_1})[V_0] - V^*\|$. By repeating the same argument we end up with $\|(\mathcal{G}_{\mathcal{N}}^{\widehat{\Pi}_k} \circ \mathcal{G}_{\mathcal{N}}^{\widehat{\Pi}_{k-1}} \circ \ldots \mathcal{G}_{\mathcal{N}}^{\widehat{\Pi}_1})[V_0] - V^*\| \leq \gamma^k \|V_0 - V^*\|$ from which the statement follows.

$\square$

**Theorem 2** *For any $\alpha$, any $\widehat{\Pi}$, and any $\mathcal{N}$, we have $\widetilde{\mathcal{G}}_{\mathcal{N}}^{\widehat{\Pi}} Q^* = Q^*$ and $(\forall Q \in l_\infty(\mathcal{S} \times \mathcal{A}))$ : $\|\widetilde{\mathcal{G}}_{\mathcal{N}}^{\widehat{\Pi}} Q - Q^*\| \leq \gamma \|Q - Q^*\|.$*

*Proof:*

$$\widetilde{\mathcal{G}}_{\mathcal{N}}^{\widehat{\Pi}} Q\left(s_0, a_0\right) \triangleq \underset{\pi \in \widehat{\Pi}}{smax^\alpha} \underset{n' \in \mathcal{N}}{smax^\alpha} \max_{n \in \{0, n'\}} \mathbb{E}_{\tau_{s_0, a_0}^{n+1} \sim \pi} \left[G_Q^{n+1}(\tau_{s_0, a_0}^{n+1})\right], \quad (14)$$

This operator can be represented as

$$\widetilde{\mathcal{G}}_{\mathcal{N}}^{\widehat{\Pi}} Q \triangleq \underset{\pi \in \widehat{\Pi}}{smax^\alpha} \underset{n' \in \mathcal{N}}{smax^\alpha} \max_{n \in \{0, n'\}} \left(\mathcal{B}^\pi\right)^n \mathcal{B} Q.$$

First, following the proof in Theorem 1, we have $\left(\mathcal{B}^\pi\right)^n \mathcal{B} Q^* \leq Q^*$, for any $\pi$ and $n \geq 0$ (with equality when $n = 0$). Then, we have

$$\underset{\pi \in \widehat{\Pi}}{smax^\alpha} \underset{n' \in \mathcal{N}}{smax^\alpha} \max_{n \in \{0, n'\}} \left(\mathcal{B}^\pi\right)^n \mathcal{B} Q^* = Q^*$$

Given an action VF $Q$, let us define two distribution over $\widehat{\Pi}$ and $\mathcal{N}$, denoted by $\mathcal{P}_{\widehat{\Pi}}^{s,a}$ and $\mathcal{P}_{\mathcal{N}}^{s,a}$ such that

$$\mathbb{E}_{\pi \sim \mathcal{P}_{\widehat{\Pi}}^{s,a}, n' \sim \mathcal{P}_{\mathcal{N}}^{s,a}} \max_{n \in \{0, n'\}} \left(\mathcal{B}^\pi\right)^n \mathcal{B} Q\left(s, a\right) = \underset{\pi \in \widehat{\Pi}}{smax^\alpha} \underset{n' \in \mathcal{N}}{smax^\alpha} \max_{n \in \{0, n'\}} \left(\mathcal{B}^\pi\right)^n \mathcal{B} Q\left(s, a\right)$$

It follows

$$\begin{aligned}
\left\|\widetilde{\mathcal{G}}_{\mathcal{N}}^{\widehat{\Pi}} Q - Q^*\right\| &= \left\|\underset{\pi \in \widehat{\Pi}}{smax^\alpha} \underset{n' \in \mathcal{N}}{smax^\alpha} \max_{n \in \{0, n'\}} \left(\mathcal{B}^\pi\right)^n \mathcal{B} Q - Q^*\right\| \\
&= \left\|\mathbb{E}_{\pi \sim \mathcal{P}_{\widehat{\Pi}}, n' \sim \mathcal{P}_{\mathcal{N}}} \max_{n \in \{0, n'\}} \left(\mathcal{B}^\pi\right)^n \mathcal{B} Q - \mathbb{E}_{\pi \sim \mathcal{P}_{\widehat{\Pi}}, n' \sim \mathcal{P}_{\mathcal{N}}} \max_{n \in \{0, n'\}} \left(\mathcal{B}^\pi\right)^n \mathcal{B} Q^*\right\| \\
&\leq \max_{\pi \in \widehat{\Pi}} \max_{n \in \mathcal{N}} \left\|\left(\mathcal{B}^\pi\right)^n \mathcal{B} Q - \left(\mathcal{B}^\pi\right)^n \mathcal{B} Q^*\right\| \\
&\leq \gamma \left\|Q - Q^*\right\|.
\end{aligned}$$

$\square$

**Corollary 1** *Assume $Q_0 \in l_\infty(\mathcal{S} \times \mathcal{A})$ and $\Pi_k, k \in \mathbb{N}$ be a sequence of sets of behavioral policies then the sequence $(\widetilde{\mathcal{G}}_{\mathcal{N}}^{\widehat{\Pi}_k} \circ \widetilde{\mathcal{G}}_{\mathcal{N}}^{\widehat{\Pi}_{k-1}} \circ \ldots \circ \widetilde{\mathcal{G}}_{\mathcal{N}}^{\widehat{\Pi}_1})[Q_0], k \in \mathbb{N}$ converges R-linearly to $Q^*$ with convergence rate $\gamma$.*

*Proof:* The proof follows from just proved point Theorem 2 and proceeds analogically to proof of Theorem 1 point 3).

$\square$

**Theorem 3** *(Comparison to Bellman Optimality Operator $\mathcal{B}$) For all $V, V_0 \in l_\infty(\mathcal{S})$ holds:*

*1)* $\left\| \mathcal{G}_{\mathcal{N}}^{\widehat{\Pi}} V - V^* \right\| \leq \|\mathcal{B}V - V^*\|$;

*2) Assume $V_0 \leq V^*$. For any $s$ we have $\left| \mathcal{G}_{\mathcal{N}}^{\widehat{\Pi}} V_0(s) - V^*(s) \right| \leq |\mathcal{B}V_0(s) - V^*(s)|$, where the strict inequality holds as long as there exists $\pi' \in \widehat{\Pi}$ such that $\arg\max_{n \in \mathcal{N}} \left( \mathcal{B}^{\pi'} \right)^n \mathcal{B}V_0(s) > 0$.*

*Proof:* 1) has been proved in Theorem 1.

2) Note that $\mathcal{G}_{\mathcal{N}}^{\widehat{\Pi}} V_0 \geq \mathcal{B}V_0$. Further, if $V_0 \leq V^*$, then $\mathcal{G}_{\mathcal{N}}^{\widehat{\Pi}} V_0 \leq V^*$ and $\mathcal{B}V_0 \leq V^*$ (as they are both monotonic). Putting all together we obtain

$$0 \leq V^* - \mathcal{G}_{\mathcal{N}}^{\widehat{\Pi}} V_0 \leq V^* - \mathcal{B}V_0,$$

from which the desired inequality follows. $\qquad\square$

**Theorem 4** *(Comparison to Multi-Step Bellman Expectation Operator) Assume that the VF $V_k \leq V^*$. Let $V_{k+1}^{\mathcal{G}_{\mathcal{N}}^{\widehat{\Pi}_k}}$ and $V_{k+1}^{\mathcal{B}_{N}^{\pi_k}}$ denote the $k+1$-th VF of Highway Value Iteration with hyperparameter $\mathcal{N}$ and Generalized Policy Iteration with hyperparameter $N$. We have*

$$\left\| V_{k+1}^{\mathcal{G}_{\mathcal{N}}^{\widehat{\Pi}_k}} - V^* \right\| \leq \min_{N \in \mathcal{N}} \left\| V_{k+1}^{\mathcal{B}_{N+1}^{\pi_k}} - V^* \right\| \tag{15}$$

*Proof:* First, for any $N \in \mathcal{N}$, $\pi' \in \widehat{\Pi}$, and any $s \in \mathcal{S}$,

$$\max_{\pi \in \widehat{\Pi}} \max_{n \in \mathcal{N}} \left( \mathcal{B}^\pi \right)^n \mathcal{B}V_k(s) \geq \left( \mathcal{B}^{\pi'} \right)^N \mathcal{B}V_k(s)$$

According to Algorithm B.1, we know that $\pi_k \in \widehat{\Pi}_k$ Then, for any s we have

$$\begin{aligned}
&\left| \mathcal{G}_{\mathcal{N}}^{\widehat{\Pi}} V_k(s) - V^*(s) \right| \\
&= \left| \max_{\pi \in \widehat{\Pi}} \max_{n \in \mathcal{N}} \left( \mathcal{B}^\pi \right)^n \mathcal{B}V_k(s) - V^*(s) \right| \\
&= V^*(s) - \max_{\pi \in \widehat{\Pi}} \max_{n \in \mathcal{N}} \left( \mathcal{B}^\pi \right)^n \mathcal{B}V_k(s) \\
&= \min_{\pi \in \widehat{\Pi}} \min_{n \in \mathcal{N}} \left( V^*(s) - \left( \mathcal{B}^\pi \right)^n \mathcal{B}V_k(s) \right) \\
&= \min_{\pi \in \widehat{\Pi}} \min_{n \in \mathcal{N}} \left| \left( \mathcal{B}^\pi \right)^n \mathcal{B}V_k(s) - V^*(s) \right| \\
&\leq \min_{n \in \mathcal{N}} \left| \left( \mathcal{B}^{\pi_k} \right)^n \mathcal{B}V_k(s) - V^*(s) \right| \\
&= \min_{N \in \mathcal{N}} \left| \left( \mathcal{B}^{\pi_k} \right)^{N+1} V_k(s) - V^*(s) \right| \\
&= \min_{N \in \mathcal{N}} \left| V_{k+1}^{\mathcal{B}_{N+1}^{\pi_k}} - V^*(s) \right|
\end{aligned}$$

from which the conclusion follows.

$\qquad\square$

**Assumption A$(\widehat{\Pi}, n)$** *Given set of behavioral policies $\widehat{\Pi}$, a lookahead depth $n$. Let $\mathcal{S}_{s,n}^{\pi'}$ denote the set of all possible visited states by executing policy $\pi'$ for $n$ steps from state $s$. For each $s \in \mathcal{S}$, there exists at least one policy $\pi' \in \widehat{\Pi}$ such that $\pi'(s') = \pi^*(s')$ for any $s' \in \mathcal{S}_{s,n}^{\pi'}$ where $\pi^*$ refers to an optimal policy. Note the quantification order: $(\forall s \in \mathcal{S}, \exists \pi' \in \widehat{\Pi}, \forall s' \in \mathcal{S}_{s,n}^{\pi'}) : \pi'(s') = \pi^*(s')$.*

**Theorem 5** *(Better Contraction Rate) Assume $N - 1 \in \mathcal{N}$, $\widehat{\Pi}$ satisfies Assumption $A(\widehat{\Pi}, N-1)$, and $V_0 \in l_\infty(\mathcal{S})$ and $V_0 \leq V^*$, the convergence rate of $\mathcal{G}_{\mathcal{N}}^{\widehat{\Pi}}$ is $\gamma^N$, i.e., $\|\mathcal{G}_{\mathcal{N}}^{\widehat{\Pi}} V - V^*\| \leq \gamma^N \|V - V^*\|$;*

*Proof:* 1) First, if $\widehat{\Pi}$ satisfies assumption Assumption A($\widehat{\Pi}, N-1$), that is, starting from any $s$, there exists at least one $\pi \in \widehat{\Pi}$ executing the optimal actions for $N-1$ steps, then we have

$$
\begin{aligned}
\mathcal{G}_{\mathcal{N}}^{\widehat{\Pi}} V &= \max_{\pi \in \widehat{\Pi}} \max_{n' \in \mathcal{N}} \left(\mathcal{B}^{\pi}\right)^{n'} \mathcal{B}V \\
&\geq \max_{\pi \in \widehat{\Pi}} \left(\mathcal{B}^{\pi}\right)^{N-1} \mathcal{B}V \\
&\geq \left(\mathcal{B}^{\pi^*}\right)^{N-1} \mathcal{B}V.
\end{aligned}
$$

Second, if $V \leq V^*$, then $\mathcal{G}_{\mathcal{N}}^{\widehat{\Pi}} V \leq V^*$ ($\mathcal{G}_{\mathcal{N}}^{\widehat{\Pi}}$ is monotonic) and $\left(\mathcal{B}^{\pi^*}\right)^{N-1} \mathcal{B}V \leq V^*$.

Finally, with the results above, we have

$$
\begin{aligned}
\left\| \mathcal{G}_{\mathcal{N}}^{\widehat{\Pi}} V - V^* \right\| &\leq \left\| \left(\mathcal{B}^{\pi^*}\right)^{N-1} \mathcal{B}V - V^* \right\| \\
&= \left\| \left(\mathcal{B}^{\pi^*}\right)^{N-1} \mathcal{B}V - \left(\mathcal{B}^{\pi^*}\right)^{N-1} \mathcal{B}V^* \right\| \\
&\leq \gamma^N \left\| V - V^* \right\|.
\end{aligned}
$$

$\square$

To utilize the better convergence rates obtained in theorem 5, we have to be a bit more careful than in theorem 1 point *3)* and take care of the monotony requirements in theorem 5. This is done in the following lemma.

**Lemma 1** *Assume a sequence of operators $T_k$, $k \in \mathbb{N}$ on $L_{\infty}(\mathcal{S})$, where all $T_k$ satisfy ($\forall V \in L_{\infty}(\mathcal{S}), V \leq V^*$) : $\|T_k V - V^*\| \leq \gamma' \|V - V^*\|$ with common convergence rate $\gamma'$ and common limit $V^*$. Further assume all $T_k$ are monotonic and have fixed point $V^*$. Then for any $V_0 \in l_{\infty}(\mathcal{S}), V_0 \leq V^*$ the sequence $(T_k \circ T_{k-1} \circ \ldots T_1)[V_0], k \in \mathbb{N}$ converges R-linearly to $V^*$ with convergence rate $\gamma'$.*

*Proof:* Assuming $V_0 \in l_{\infty}(\mathcal{S}), V_0 \leq V^*$ and monotony and fixed point of operators $T_1, \ldots, T_{k-1}$ implies: $V := (T_{k-1} \circ T_{k-2} \circ \ldots T_1)[V_0] \leq V^*$. Now we can apply the inequality with $\gamma'$ to get $\|(T_k \circ T_{k-1} \circ \ldots T_1)[V_0] - V^*\| \leq \gamma' \|(T_{k-1} \circ \ldots T_1)[V_0] - V^*\|$. By repeating the same argument we end up with $\|(T_k \circ T_{k-1} \circ \ldots T_1)[V_0] - V^*\| \leq \gamma'^k \|V_0 - V^*\|$ from which the statement follows.

$\square$

Let us denote $\widehat{\Pi}_k$ ($k \in \mathbb{N}$) the sequence of behavioral policies generated by Algorithm B.1. In Algorithm B.1 the set of behavioral policies $\widehat{\Pi}_k$ changes by adding a greedy policy every $K$ iterations. Regardless of the sets $\widehat{\Pi}_k$, the algorithm converges to the optimal value function form theorem 1 point *3)*. This means that more and more close to optimal policies are added to $\widehat{\Pi}_k$ set with growing iteration number $k$. When assumption Assumption A($\cdot, n$) gets satisfied for some $n$ at iteration $k_0$, then due to monotony of the sequence $\widehat{\Pi}_k, k \in \mathbb{N}$ (i.e. the set $\widehat{\Pi}_k$ can just grow over time) Assumption A($\cdot, n$) is satisfied for the whole suffix $\widehat{\Pi}_k, k \geq k_0$. We can then use lemma 1 on corresponding suffix of operator sequence $\mathcal{G}_{\mathcal{N}}^{\widehat{\Pi}_k}, k \geq k_0$ with $\gamma' = \gamma^{n+1}$ to claim better convergence rate ($\gamma^{n+1}$) of the corresponding suffix of value functions $V_k, k \geq k_0$. When $\mathcal{S}$ is finite this gives us monotonic improvement of the convergence rate of the sequence $V_k, k \in \mathbb{N}$ as Assumption A($\cdot, n$) gets eventually satisfied for bigger and bigger $n$.

**Theorem 6** *(Highway Bellman Optimality Equation (Highway Equation) w.r.t. action VF Q) For any $\widehat{\Pi}$ and $\mathcal{N}$, we have*

*1. $\mathcal{G}_{\mathcal{N}}^{\widehat{\Pi}}$ is a contraction on complete metric space $l_{\infty}(\mathcal{S} \times \mathcal{A})$, i.e. for any $Q, Q' \in l_{\infty}(\mathcal{S} \times \mathcal{A})$, we have $\|\mathcal{G}_{\mathcal{N}}^{\widehat{\Pi}} Q - \mathcal{G}_{\mathcal{N}}^{\widehat{\Pi}} Q'\| \leq \|\mathcal{B}Q - \mathcal{B}Q'\| \leq \gamma \|Q - Q'\|$;*

*2. $Q^*$ is the only fixed point $\mathcal{G}_{\mathcal{N}}^{\widehat{\Pi}}$. That is,*

$$
\mathcal{G}_{\mathcal{N}}^{\widehat{\Pi}} Q^* = Q^*, \tag{16}
$$

*and for all $Q \in l_\infty(\mathcal{S} \times \mathcal{A})$ holds $Q = \mathcal{G}_\mathcal{N}^{\widehat{\Pi}} Q$ if and only if $Q = Q^*$.*

*Proof:* Note that $\mathcal{G}_\mathcal{N}^{\widehat{\Pi}}$ w.r.t action VF $Q$ can also be represented by the Bellman Operators w.r.t $Q$, i.e.,

$$\mathcal{G}_\mathcal{N}^{\widehat{\Pi}} Q \triangleq \max_{\pi \in \widehat{\Pi}} \max_{n \in \mathcal{N}} (\mathcal{B}^\pi)^n \mathcal{B} Q.$$

Except for the change of value spaces ($l_\infty(\mathcal{S} \times \mathcal{A})$ instead of $l_\infty(\mathcal{S})$) the proof is the same as for Theorem 1). $\qquad\square$

**Lemma 2** *If $V_0 \leq V^*$, then $(\mathcal{G}_\mathcal{N}^{\widehat{\Pi}_k} \circ \mathcal{G}_\mathcal{N}^{\widehat{\Pi}_{k-1}} \circ \ldots \circ \mathcal{G}_\mathcal{N}^{\widehat{\Pi}_1}) V_0 \leq V^*$ for any $k$ and any sequence of policy sets $\widehat{\Pi}_1, \widehat{\Pi}_2, \ldots, \widehat{\Pi}_k$.*

*Proof:* As $\mathcal{G}_\mathcal{N}^{\widehat{\Pi}}$ is monotonic (for any set of behavioral policies $\widehat{\Pi}$), we have $(\mathcal{G}_\mathcal{N}^{\widehat{\Pi}_1}) V_0 \leq (\mathcal{G}_\mathcal{N}^{\widehat{\Pi}_1}) V^* = V^*$. Further, applying $\mathcal{G}_\mathcal{N}^{\widehat{\Pi}_2}$ again to the both sides we obtain $(\mathcal{G}_\mathcal{N}^{\widehat{\Pi}_2} \circ \mathcal{G}_\mathcal{N}^{\widehat{\Pi}_1}) V_0 \leq (\mathcal{G}_\mathcal{N}^{\widehat{\Pi}_2}) V^* = V^*$. Repeting the argument further $k - 2$ times we get the result.

$\qquad\square$

## A.2  Proof of properties of Multi-step Bellman Optimality operator

Recent works combined multi-step trajectory data and estimated the values of the most promising actions without any correction for the off-policy data (Horgan et al., 2018; Barth-Maron et al., 2018). The underlying operator is defined as

$$\mathcal{B}_N^{\widehat{\Pi}} Q (s_0, a_0) \triangleq \mathbb{E}_{\pi \sim \mathcal{P}_{\widehat{\Pi}}} \left[ (\mathcal{B}^\pi)^{N-1} \mathcal{B} Q(s_0, a_0) \right]$$
$$= \mathbb{E}_{\pi \sim \mathcal{P}_{\widehat{\Pi}}, \tau_{s_0, a_0}^N \sim \pi} \left[ \sum_{t=0}^{N-1} \gamma^t r_t + \gamma^N \max_{a_N'} Q (s_N, a_N') \right] \qquad (17)$$

where $\mathcal{P}_{\widehat{\Pi}}$ is a distribution over $\widehat{\Pi}$ and $\mathcal{P}_{\widehat{\Pi}}(\pi)$ is the probability of selecting $\pi$. Here we use $\pi \sim \mathcal{P}_{\widehat{\Pi}}$ and $\tau_{s_0, a_0}^N \sim \pi$ to formulate the procedure of prioritized experience replay (Schaul et al., 2015; Horgan et al., 2018), which samples the trajectory data collected by various behavioral policies according to a prior distribution. Although these methods have shown promising results in practice, below we will show that this operator is generally biased w.r.t. the optimal action VF $Q^*$. In other words, the corresponding fixed point $Q^*_{\mathcal{B}_N^{\widehat{\Pi}}}$ is different from the optimal VF $Q^*$, and unbiased learning only happens when *all* behavioral policies are optimal.

**Theorem 7** *(Properties of the Multi-step Bellman Optimality Operator $\mathcal{B}_N^{\widehat{\Pi}}$) For any $N \geq 2$,*
*1) The operator $\mathcal{B}_N^{\widehat{\Pi}}$ is a contraction on complete metric space $l_\infty(\mathcal{S} \times \mathcal{A})$, i.e., for any two vectors $Q, Q' \in l_\infty(\mathcal{S} \times \mathcal{A})$, $\|\mathcal{B}_N^{\widehat{\Pi}} Q - \mathcal{B}_N^{\widehat{\Pi}} Q'\| \leq \gamma^N \|Q - Q'\|$.*
*2) Let $Q^*_{\mathcal{B}_N^{\widehat{\Pi}}}$ denote the fixed point of $\mathcal{B}_N^{\widehat{\Pi}}$, i.e., $\mathcal{B}_N^{\widehat{\Pi}} Q^*_{\mathcal{B}_N^{\widehat{\Pi}}} = Q^*_{\mathcal{B}_N^{\widehat{\Pi}}}$, we have $Q^*_{\mathcal{B}_N^{\widehat{\Pi}}} \leq Q^*$.*
*3) $Q^*_{\mathcal{B}_N^{\widehat{\Pi}}} = Q^*$ if and only if any $\pi \in \widehat{\Pi}$, $\mathcal{P}_{\widehat{\Pi}}(\pi) > 0$ satisfies $\pi(s) = \pi^*(s)$ for any $s \in U := \{s_1 \in \mathcal{S} | \exists s_0, a_0, \mathcal{T}(s_1|s_0, a_0) > 0\}$ and $\pi^*$ an optimal policy.*

Before giving the proof of the above theorem, we'd like to mention a variant of our Highway Operator, named *Expectation Highway Operator*, defined as

$$\overline{\mathcal{G}}_{\widehat{\Pi}}^\mathcal{N} Q(s_0, a_0) \triangleq \mathbb{E}_{\pi \sim \mathcal{P}_{\widehat{\Pi}}} \left[ \max_{n \in \mathcal{N}} (\mathcal{B}^\pi)^{n-1} \mathcal{B} Q(s_0, a_0) \right] \qquad (18)$$

Compared to the above Multi-Step Bellman Optimality Operator in eq. (17), our operator uses maximization over lookahead depths instead of a fixed lookahead depth. It's interesting to note that this operator is unbiased w.r.t. $Q^*$ for any set of behavioral policies $\widehat{\Pi}$. Formally, we have the following theorem.

**Theorem 8** *For any $\widehat{\Pi}$, any $\mathcal{P}_{\widehat{\Pi}}$, and any $\mathcal{N}$,*

*1) $\overline{\mathcal{G}}_{\widehat{\Pi}}^{\mathcal{N}}$ is a contraction on complete metric space $l_\infty(\mathcal{S} \times \mathcal{A})$, i.e., for any two vectors $Q, Q' \in l_\infty(\mathcal{S} \times \mathcal{A})$, $(\forall Q \in l_\infty(\mathcal{S} \times \mathcal{A})) : \|\overline{\mathcal{G}}_{\widehat{\Pi}}^{\mathcal{N}} Q - \overline{\mathcal{G}}_{\widehat{\Pi}}^{\mathcal{N}} Q'\| \leq \gamma \|Q - Q'\|$;*
*2) $\overline{\mathcal{G}}_{\widehat{\Pi}}^{\mathcal{N}} Q^* = Q^*$.*

We now give the proofs of Theorem 7 and 8 respectively.

*Proof of Theorem 7:*

For simplicity, we will use $\mathcal{P}$ instead of $\mathcal{P}_{\widehat{\Pi}}$.

1)

$$\|\mathcal{B}_N^{\widehat{\Pi}} Q - \mathcal{B}_N^{\widehat{\Pi}} Q'\| \leq \mathbb{E}_{\pi \sim \mathcal{P}} \left\| \left(\mathcal{B}^\pi\right)^{N-1} \mathcal{B}Q - \left(\mathcal{B}^\pi\right)^{N-1} \mathcal{B}Q' \right\|$$
$$\leq \gamma^N \|Q - Q'\|$$

2) Applying Banach's fixed point theorem to $\mathcal{B}_N^{\widehat{\Pi}}$ using the contraction result, we know that $\mathcal{B}_N^{\widehat{\Pi}}$ has only one fixed point.

Following the proof in Theorem 1, we have $(\mathcal{B}^\pi)^{N-1}\mathcal{B}Q^* \leq Q^*$ for any $\pi$ and $N \geq 2$. This implies that

$$Q^* \geq \mathbb{E}_\pi (\mathcal{B}^\pi)^{N-1} \mathcal{B}Q^* = \mathcal{B}_N^{\widehat{\Pi}} Q^*. \tag{19}$$

Using monotony of $\mathcal{B}_N^{\widehat{\Pi}}$ ($Q \leq Q'$ implies $\mathcal{B}_N^{\widehat{\Pi}} Q \leq \mathcal{B}_N^{\widehat{\Pi}} Q'$), we get the monotonic sequence $Q^* \geq \mathcal{B}_N^{\widehat{\Pi}} Q^* \geq (\mathcal{B}_N^{\widehat{\Pi}})^2 Q^* \geq \cdots$ which converges to the fixed point $Q^*_{\mathcal{B}_N^{\widehat{\Pi}}}$ based on the contraction property and Banach fixed point theorem :

$$Q^* \geq \mathcal{B}_N^{\widehat{\Pi}} Q^* \geq (\mathcal{B}_N^{\widehat{\Pi}})^2 Q^* \geq \cdots \geq (\mathcal{B}_N^{\widehat{\Pi}})^k Q^* \searrow Q^*_{\mathcal{B}_N^{\widehat{\Pi}}}. \tag{20}$$

3) For the implication "$\Longleftarrow$"it suffices to show for all $\pi \in \widehat{\Pi}$, $\mathcal{P}(\pi) > 0$ $\mathcal{B}^\pi Q^* = Q^*$. Since then we get

$$\mathcal{B}_N^{\widehat{\Pi}} Q^* = \mathbb{E}_{\pi \sim \mathcal{P}} (\mathcal{B}^\pi)^{N-1} Q^* = \mathbb{E}_{\pi \sim \mathcal{P}} Q^* = Q^*$$

and the implication is proved. Thus let us fix $\pi \in \widehat{\Pi}$, $\mathcal{P}(\pi) > 0$ and $s_0 \in \mathcal{S}, a_0 \in \mathcal{A}$ the following holds $\mathcal{B}^\pi Q^*(s_0, a_0) = \mathbb{E}_{s_1 \sim \mathcal{T}(\cdot|s_0, a_0)} \mathbb{E}_{a_1 \sim \pi(\cdot|s_1)}[r(s_0, a_0) + \gamma Q^*(s_1, a_1)]$. Since in the first expectation we just care about $s_1$ for which $\mathcal{T}(s_1|s_0, a_0) > 0$, we can assume $s_1 \in U$. As $\pi$ on $U$ can be replaced by the optimal policy $\pi^*$ from the assumption, we get

$$\mathcal{B}^\pi Q^*(s_0, a_0) = \mathbb{E}_{s_1 \sim \mathcal{T}(\cdot|s_0, a_0)} \mathbb{E}_{a_1 \sim \pi(\cdot|s_1)}[r(s_0, a_0) + \gamma Q^*(s_1, a_1)]$$
$$= \mathbb{E}_{s_1 \sim \mathcal{T}(\cdot|s_0, a_0)} \mathbb{E}_{a_1 \sim \pi^*(\cdot|s_1)}[r(s_0, a_0) + \gamma Q^*(s_1, a_1)] = Q^*(s_0, a_0).$$

The remaining implication "$\Longrightarrow$" will be proved by contradiction. Assume that the conclusion does not hold, i.e. there exist $\pi \in \widehat{\Pi}$, $\mathcal{P}(\pi) > 0$ and $s_1 \in U$ such that $\mathcal{P}(\pi) > 0$ and $\pi(\cdot|s_1)$ is not optimal. Since $s_1 \in U$ there exists $s_0 \in \mathcal{S}, a_0 \in \mathcal{A}$ such that $\mathcal{T}(s_1|s_0, a_0) > 0$. First we aim to prove inequality

$$\mathcal{B}^\pi Q^*(s_0, a_0) < Q^*(s_0, a_0).$$

Since $\pi(\cdot|s_1)$ assigns positive probability to non-optimal action, it is easy to obtain (especially for finite $\mathcal{A}$) that

$$\mathbb{E}_{a_1 \sim \pi(\cdot|s_1)} Q^*(s_1, a_1) < V^*(s_1).$$

For other states different from $s_1$ we can still have equality but the countable sum leaves the inequality strict

$$(\mathcal{B}^\pi Q^*)(s_0, a_0) = r(s_0, a_0) + \mathbb{E}_{s_1' \sim \mathcal{T}(\cdot|s_0, a_0)} \mathbb{E}_{a_1 \sim \pi(\cdot|s_1')} Q^*(s_1', a_1)$$
$$< r(s_0, a_0) + \mathbb{E}_{s_1' \sim \mathcal{T}(\cdot|s_0, a_0)} V^*(s_1') = Q^*(s_0, a_0).$$

Now since $(\mathcal{B}^\pi)^{N-2} Q^* \leq Q^*$ we get

$$(\mathcal{B}^\pi)^{N-1} \mathcal{B}Q^* = (\mathcal{B}^\pi)^{N-1} Q^* = \mathcal{B}^\pi (\mathcal{B}^\pi)^{N-2} Q^* \leq \mathcal{B}^\pi Q^*.$$

Using previous result we obtain

$$((\mathcal{B}^\pi)^{N-1}\mathcal{B}Q^*)(s_0, a_0) \leq \mathcal{B}^\pi Q^*(s_0, a_0) < Q^*(s_0, a_0),$$

and applying expectation with finite set $\widehat{\Pi}$ using $\mathcal{P}(\pi) > 0$ we get

$$\mathcal{B}_N^{\widehat{\Pi}} Q^*(s_0, a_0) < Q^*(s_0, a_0),$$

which can be combined with eq. (20) to show

$$Q^*_{\mathcal{B}_N^{\widehat{\Pi}}}(s_0, a_0) < Q^*(s_0, a_0).$$

$\square$

We now give proof of Theorem 8.

*Proof of Theorem 8:* 1) We first prove the contraction property,

$$
\begin{aligned}
\|\overline{\mathcal{G}}_{\widehat{\Pi}}^{\mathcal{N}} Q - \overline{\mathcal{G}}_{\widehat{\Pi}}^{\mathcal{N}} Q'\| &\leq \left\| \mathbb{E}_{\pi \sim \mathcal{P}_{\widehat{\Pi}}} \left[ \max_{n \in \mathcal{N}} (\mathcal{B}^\pi)^n \mathcal{B}Q \right] - \mathbb{E}_{\pi \sim \mathcal{P}_{\widehat{\Pi}}} \left[ \max_{n \in \mathcal{N}} (\mathcal{B}^\pi)^n \mathcal{B}Q' \right] \right\| \\
&\leq \mathbb{E}_{\pi \sim \mathcal{P}_{\widehat{\Pi}}} \max_{n \in \mathcal{N}} \|(\mathcal{B}^\pi)^n \mathcal{B}Q - (\mathcal{B}^\pi)^n \mathcal{B}Q'\| \\
&\leq \gamma \|Q - Q'\|
\end{aligned}
$$

2) Similar to the proof in Theorem 1, $(\mathcal{B}^\pi)^n \mathcal{B}Q^* \leq Q^*$ for any $\pi$ and $n \geq 0$ (and the equality holds when $n = 0$), we have

$$\overline{\mathcal{G}}_{\widehat{\Pi}}^{\mathcal{N}} Q^* \triangleq \mathbb{E}_{\pi \sim \mathcal{P}_{\widehat{\Pi}}} \left[ \max_{n \in \mathcal{N}} (\mathcal{B}^\pi)^n \mathcal{B}Q^* \right] = Q^*$$

$\square$

## A.3 DISCUSSION ABOUT SOFTMAX HIGHWAY OPERATOR

The Softmax Highway Operator is defined by

$$\widetilde{\mathcal{G}}_{\mathcal{N}}^{\widehat{\Pi}} Q \triangleq \underset{\pi \in \widehat{\Pi}}{smax^\alpha} \underset{n' \in \mathcal{N}}{smax^\alpha} \max_{n \in \{0, n'\}} (\mathcal{B}^\pi)^n \mathcal{B}Q,$$

If we remove the $\max_{n \in \{0, n'\}}$ in the above operator, i.e.,

$$\underset{\pi \in \widehat{\Pi}}{smax^\alpha} \underset{n' \in \mathcal{N}}{smax^\alpha} (\mathcal{B}^\pi)^n \mathcal{B}Q$$

then the above operator is biased w.r.t. $Q^*$. This operator can be regarded as an extension to the multi-step Bellman Optimality Operator $\mathcal{B}_n^{\widehat{\Pi}}$, average over various $n$ with weights $\underset{n' \in \mathcal{N}}{smax^\alpha}(\cdot)$. Therefore, it has similar biased property to $\mathcal{B}_n^{\widehat{\Pi}}$. However, simply adding $\max_{n \in \{0, n'\}}$ with minor computational cost, our Softmax Highway operator is unbiased while alleviating the overestimation issue and improving the exploration.

## A.4 MULTI-STEP IMPORTANCE SAMPLING-BASED BELLMAN EXPECTATION OPERATOR

In this section, we describe the classical off-policy learning method based on importance sampling (IS). IS-based off-policy methods evaluate the value function of a policy $\pi'$ (called *target policy*) using the data collected by a different policy $\pi$ (called *behavior policy*). The underlying operator, called *Importance Sampling-based Bellman Expectation Operator* (Sutton & Barto, 2018), is defined as follows,

$$\breve{\mathcal{B}}_\pi^{\pi'} Q(s, a) \triangleq \mathbb{E}_{s' \sim \mathcal{T}(\cdot|s,a), a' \sim \pi(\cdot|s')} \left[ \frac{\pi'(a'|s')}{\pi(a'|s')} (r(s, a) + \gamma Q(s', a')) \right] \tag{21}$$

To utilize the multi-step data collected by different behavior policies, the above operator can be extended to a multi-step version with a set of behavioral policies (we call *Multi-Step IS-based Bellman Expectation Operator*) (Sutton & Barto, 2018), which is defined as

$$(\mathcal{B}_N^{\pi,\widehat{\Pi}}Q)(s_0, a_0) \triangleq \mathbb{E}_{\pi \sim \mathcal{P}_{\widehat{\Pi}}}\left[ (\breve{\mathcal{B}}_\pi^{\pi'})^N Q(s_0, a_0) \right]$$

$$= \mathbb{E}_{\pi \sim \mathcal{P}_{\widehat{\Pi}}, \tau_{s_0}^N \sim \pi}\left[ \sum_{t=0}^{N-1} \gamma^t \zeta^{1:t} r_t + \gamma^N \zeta^{1:N} Q(s_N, a_N) \right] \tag{22}$$

where $N$ is the lookahead depth/bootstrapping step; $\tau_{s_0}^N = (s_0, a_0, s_1, a_1, s_2, a_2, \cdots, s_N)$; $\tau_{s_0}^N \sim \pi$ is the trajectory starting from $s_0$ by executing policy $\pi$ for $N$ steps; and $\zeta^{1:t} \triangleq \prod_{t'=1}^t \frac{\pi'(a_{t'}|s_{t'})}{\pi(a_{t'}|s_{t'})}$ is the product of IS ratios. The products of IS ratios could cause high variance.

## B    METHOD

### B.1    ALGORITHMS

#### B.1.1    HIGHWAY VALUE ITERATION

The Highway Value Iteration algorithm is presented as Algorithm B.1.

#### B.1.2    HIGHWAY Q-LEARNING

The Highway Q-Learning algorithm is presented as Algorithm B.3.

#### B.1.3    HIGHWAY DQN

The Highway DQN algorithm is presented as Algorithm B.4.

## C    EXPERIMENTAL RESULTS

The code of this paper is publicly-available at `https://anonymous.4open.science/r/Highway-Reinforcement-Learning-4202`.

### C.1    EXPERIMENTS WITH MODEL-BASED ALGORITHMS

#### C.1.1    DETAILS OF ENVIRONMENTS

**Multi-Room** is a grid world environment with multiple rooms connected by doors. The agent's goal is to reach a goal square in the opposite corner and get a reward ($r = 1000$). In addition, the agent will get a small reward $r = 0.001$ when it finds the exit door of the room. We use the implementations based on the gym-minigrid (Chevalier-Boisvert et al., 2018).

---

**Algorithm B.1** Highway Value Iteration

---

**Input:** Initial set of behavioral policies $\widehat{\Pi}_0$; the set of lookahead depths $\mathcal{N}$; interval for adding new policy $K$ .
**Initialize:** Initial VF $V_0 \in \mathbb{R}^{|\mathcal{S}|}$, $\epsilon$.
**for** $k = 1, 2, \ldots$ **do**
  **if** $(k-1) \bmod K == 0$ **then**
    $\pi_k(s) = \arg\max_a [r(s,a) + \gamma \mathbb{E}_{s'}[V_{k-1}(s')]]$
    $\widehat{\Pi}_k = \widehat{\Pi}_{k-1} \cup \{\pi_k\}$
  **end if**
  $V_k \leftarrow \mathcal{G}_{\mathcal{N}}^{\widehat{\Pi}_k} V_{k-1}$
  **if** $\|V_k - V_{k-1}\|_\infty \leq \epsilon$ **then break end if**
**end for**

---

---

**Algorithm B.2** Generalized Policy Iteration (Sutton & Barto, 2018)

---

**Input:** the lookahead depth $N$.
**Initialize:** Initial VF $V_0 \in \mathbb{R}^{|\mathcal{S}|}$, $\epsilon$.
**for** $k = 1, 2 \ldots$ **do**
$\quad \pi_k(s) = \arg\max_a [r(s, a) + \gamma \mathbb{E}_{s'}[V_{k-1}(s')]]$
$\quad V_k \leftarrow (\mathcal{B}^{\pi_k})^{\circ N} V_{k-1}$
$\quad$ **if** $\|V_k - V_{k-1}\|_\infty \leq \epsilon$ **then break end if**
**end for**

---

---

**Algorithm B.3** Highway Q-Learning

---

1: **Input:** Set of lookahead depths $\mathcal{N}$; Number of behavior policies $M$ (for computing target Q value); Epochs of running algorithm $I_{\mathrm{run}}$; Number of behavior policies $M$ (for computing target Q value); Number of searched behavior policies $M$; Epochs of rolling-out policy $I_{\mathrm{rollout}}$; Epochs of updating value function $I_{\mathrm{update}}$; Initial value function $Q_0 \in \mathbb{R}^{|\mathcal{S}| \times |\mathcal{A}|}$; Exploration rate $\epsilon$.
2: **Initialize:** $k = 0$; State-action replay buffer $\mathcal{D} = \emptyset$; .
3: **for** $m = 1, \cdots, I_{\mathrm{run}}$ **do**
4: $\quad$ Set $\pi_m$ to be $\epsilon$-greedy policy with $Q_k$
5: $\quad \mathcal{D}_{s,a}^{(m)} \leftarrow \emptyset$, for all $(s, a) \in \mathcal{S} \times \mathcal{A}$
6: $\quad$ **for** $j = 1, \cdots, I_{\mathrm{rollout}}$ **do**
7: $\quad\quad$ Collect a trajectory $\tau = (s_0, a_0, r_0, s_1, a_1, r_1, \cdots, s_T)$ with $\pi_m$.
8: $\quad\quad$ **for** $t = 0, 1, \cdots, T - 1$ **do**
9: $\quad\quad\quad$ Add $(s_t, a_t, r_t, s_{t+1}, a_{t+1}, r_{t+1}, \cdots, s_T)$ to $\mathcal{D}_{s_t, a_t}^{(m)}$
10: $\quad\quad\quad$ Add $(s_t, a_t)$ to $\mathcal{D}$
11: $\quad\quad$ **end for**
12: $\quad$ **end for**
13: $\quad$ **for** $j = 1, \cdots, I_{\mathrm{update}}$ **do**
14: $\quad\quad$ Sample a $(s_0, a_0)$ from $\mathcal{D}$.
15: $\quad\quad$ Update the value function by the following rules

$$Q_{k+1}(s_0, a_0) = \max_{m \in \mathsf{M}_{s_0, a_0}} \max_{n \in \mathcal{N}} \widehat{\mathbb{E}}^{\mathcal{D}_{s_0, a_0}^{(m)}} \left[ G_{Q_k}^{n+1}(\tau_{s_0, a_0}^{n+1}) \right]$$

$$\text{where} \quad \widehat{\mathbb{E}}^{\mathcal{D}_{s_0, a_0}^{(m)}} [\cdot] = \frac{1}{\mathcal{D}_{s,a}^{(m)}} \sum_{r_{s_0, a_0}^{n+1} \in \mathcal{D}_{s_0, a_0}^{(m)}} [\cdot]$$

$$\mathsf{M}_{s_0, a_0} = \{m_1, \cdots, m_M\}, m_i \sim Uniform\left( \left\{ m' \big| |\mathcal{D}_{s_0, a_0}^{(m')}| \neq 0 \right\} \right)$$

16: $\quad\quad k = k + 1$
17: $\quad$ **end for**
18: **end for**

---

### C.1.2 DETAILS OF ALGORITHMS

We compare our Highway Value Iteration to Policy Iteration and Value Iteration. For Policy Iteration, the lookahead step is set by $N = 10$ (see Algorithm B.2). For our Highway Value Iteration method, the interval of adding policy is set by $K = 7$; the set of lookahead depths $\mathcal{N} = \{0, 1, 2, \cdots, 9\}$; the size of behavioral policies is set by $|\widehat{\Pi}| = 5$. The error bound ($\|V_k - V_{k-1}\|_\infty \leq \epsilon$) for all algorithms is set by $\epsilon = 10^{-10}$.

### C.2 EXPERIMENTS OF MODEL-FREE ALGORITHMS IN TOY TASKS

Here we present the details of model-free-algorithms in artificial tasks. We adopt the experimental setting of Arjona-Medina et al. (2019).

---

**Algorithm B.4** Highway DQN

---

**Input:** Set of lookahead depths $\mathcal{N}$; Number of behavior policies $M$ (for computing target Q value); Epochs of running algorithm $I_{\mathrm{run}}$; Epochs of rolling-out policy $I_{\mathrm{rollout}}$; Epochs of updating value function $I_{\mathrm{update}}$; Batch size of updating value function $b$; Initial parameter $\theta$ of $Q$ function; Exploration rate $\epsilon$; Temperature hyperparameter $\alpha$ for softmax function $smax^\alpha$.

**Initialize:** State-action replay buffer $\mathcal{D} = \emptyset$; Initialize parameter $\theta'$ of target $Q$ function, $\theta' \leftarrow \theta$.

**for** $m = 1, \cdots, I_{\mathrm{run}}$ **do**

    Set $\pi_m$ to be $\epsilon$-greedy policy with $Q_\theta$

    $\mathcal{D}_{s,a}^{(m)} \leftarrow \emptyset$, for all $(s,a) \in \mathcal{S} \times \mathcal{A}$

    **for** $j = 1, \cdots, I_{\mathrm{rollout}}$ **do**

        Collect a trajectory $\tau = (s_0, a_0, r_0, s_1, a_1, r_1, \cdots, s_T)$ with $\pi_m$

        **for** $t = 0, 1, \cdots, T-1$ **do**

            Add $(s_t, a_t, r_t, s_{t+1}, a_{t+1}, r_{t+1}, s_{t+2}, a_{t+2}, \cdots, s_T)$ to $\mathcal{D}_{s_t, a_t}^{(m)}$

            Add $(s_t, a_t)$ to $\mathcal{D}$

        **end for**

    **end for**

    **for** $j = 1, \cdots, I_{\mathrm{update}}$ **do**

        Sample a mini-batch $\widehat{\mathcal{D}} = \{(s,a)\}$ from $\mathcal{D}$.

        Update $\theta$ by minimizing the following loss function:

$$L(\theta) = \sum_{(s,a)\in\widehat{\mathcal{D}}} \left[ Q_\theta(s,a) - \underset{m\in\mathsf{M}_{s,a}}{smax^\alpha}\underset{n'\in\mathcal{N}}{smax^\alpha} \max_{n\in\{0,n'\}} \widehat{\mathbb{E}}^{\mathcal{D}_{s,a}^{(m)}}\left[G_{Q_{\theta'}}^{n+1}(\tau_{s,a}^{n+1})\right] \right]^2 \tag{23}$$

        where $\quad \widehat{\mathbb{E}}^{\mathcal{D}_{s_0,a_0}^{(m)}}[\cdot] = \dfrac{1}{\mathcal{D}_{s,a}^{(m)}} \displaystyle\sum_{r_{s_0,a_0}^{n+1}\in\mathcal{D}_{s_0,a_0}^{(m)}} [\cdot]$

$$\mathsf{M}_{s_0,a_0} = \{m_1, \cdots, m_M\}, m_i \sim Uniform\left(\left\{m' \big| |\mathcal{D}_{s_0,a_0}^{(m')}| \neq 0\right\}\right)$$

    **end for**

    $\theta' \leftarrow \theta$ every $C$ timesteps

**end for**

---

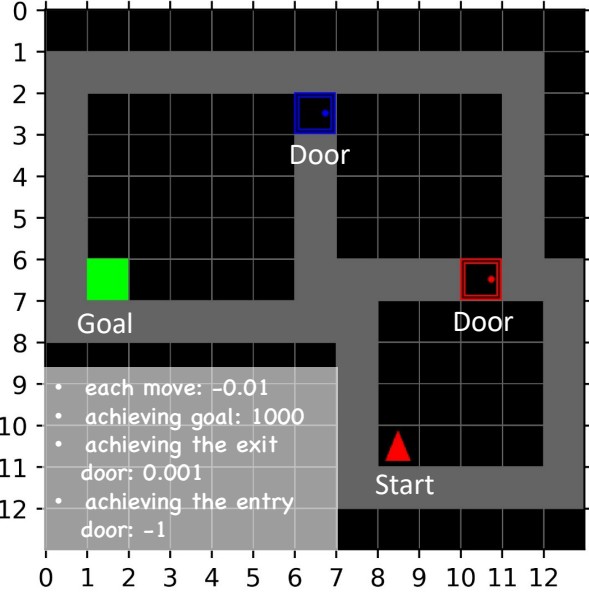

Figure 5: The illustration of the Minimalistic Gridworld Environment

### C.2.1 DETAILS OF ENVIRONMENTS

We evaluate the model-free algorithms on two toy tasks involving delayed rewards (Arjona-Medina et al., 2019), where a reward is only provided at the end of each trial and is associated with the previous actions. For example, in task "Trace Back," the final reward depends on the first two actions. Each task is run with 100 random seeds. In task "Choice," the stochastic reward depends on the first action at the beginning; the final reward depends on the first two actions. Please refer to Arjona-Medina et al. (2019) for more details.

### C.2.2 ALGORITHMIC DETAILS

The following methods are compared:

- RUDDER with reward redistribution for $Q$-value estimation, and RUDDER applied on top of $Q$-learning.

- $Q$-learning with eligibility traces according to Watkins ($Q(\lambda)$).

- SARSA with eligibility traces (SARSA($\lambda$)).

- Monte Carlo.

For RUDDER, we use the default setting of Arjona-Medina et al. (2019). For Q($\lambda$) and SARSA($\lambda$), the hyperparameter of eligibility traces is $\lambda = 0.9$. For Q($\lambda$), we use Watkins' implementation.

The algorithms are evaluated until the task is solved. For MC, Q-values are the exponential moving average of the episode return. In all experiments, an $\epsilon$-greedy policy with $\epsilon = 0.2$ is adopted.

For our Highway Q-Learning, the set of lookahead depths $\mathcal{N}$ is set by $\{0, 1, 2, \cdots, N - 1\}$, where $N = T - t$ is set to be the length until the end ($T$ is the length of trajectory and $t$ the timestep of current state). Epochs of rolling-out policy is set by $I_{\text{rollout}} = 1$. Epochs of updating value function is set by $I_{\text{update}} = T$.

## C.3 EXPERIMENTS OF MODEL-FREE ALGORITHMS IN MINATAR TASKS

### C.3.1 ENVIRONMENTAL DETAILS

This paper uses 5 games in MinAtar (Young & Tian, 2019), including Asterix, Breakout, Freeway, Seaquest, and Space Invaders. The details of the environments can be found at `https://github.com/kenjyoung/MinAtar`.

### C.3.2 DETAILS OF ALGORITHMS

We implement our and competing methods by extending the implementation of Maxmin DQN Lan (2019). The hyperparameters of training deep neural network are provided in Table 2.

We reuse the hyper-parameters and settings of neural networks in the Maxmin DQN paper (Lan et al., 2020). For Maxmin DQN, the best number of target networks (a hyperparameter) was chosen from $[2, 3, 4, 5, 6, 7, 8, 9]$ and the best learning rates were chosen from $[3 \times 10^{-3}; 3 \times 10^{-4}; 3 \times 10^{-5}]$. The set of lookahead depths of our Highway DQN is set by $\{0, 1, \cdots, N - 1\}$, and the lookahead depth/bootstrapping step of Multi-Step DQN and Retrace($\lambda$) is set by $N$. For our Highway DQN, Multi-Step DQN and Multi-Step SARSA, the best hyperparameter $N$ was chosen from $[4, 8, 12]$; the best learning rate from $[3 \times 10^{-3}; 3 \times 10^{-4}]$, and the best number of target networks was chosen from $[2, 4, 6]$. For Retrace($\lambda$), $\lambda$ is set by 1 according to the suggestion of Retrace($\lambda$) paper (Munos et al., 2016). For our Highway DQN, the best hyperparameter of softmax temperature $\alpha$ was chosen from $\{0.1, 0.5\}$. Epochs of rolling-out policy is set by $I_{\text{rollout}} = 1$.

In practice, we adopt several measures to accelerate the computation of Highway DQN. We cache the Q values generated by the target network for data $(s, a)$ such that they can be reused when the data is sampled again for training until the target network is updated. With the cached Q values, all we need is a softmax over $|\mathcal{N}| \times |\mathcal{K}|$ numbers, which is typically fast on GPUs.

| Hyperparameter | Value |
|---|---|
| Optimizer | RMSprop |
| Batch size | 32 |
| Gradient Clipping | Gym:5
MinAtari:1 |
| Target network update frequency | Gym:200
MinAtari:1000 |
| Hidden units of Q network | Gym: Fully-connected layer (64,64)
MinAtari:
    Conv. layer(out channels=16, kernel size=3, stride=1)
    Conv. layer(out channels=16, kernel size=3, stride=1) |
| Activation function | ReLU |
| Buffer size | Gym:$10^4$
MinAtari:$10^5$ |
| Discount factor $\gamma$ | 0.99 |
| Exploration rate $\epsilon$ | statr:1.0; end: 0.1; decay type: linear |

Table 2: Hyperparameters of the implemented algorithms. We reused hyperparameters and settings of neural networks in the Maxmin DQN paper (Lan et al., 2020).

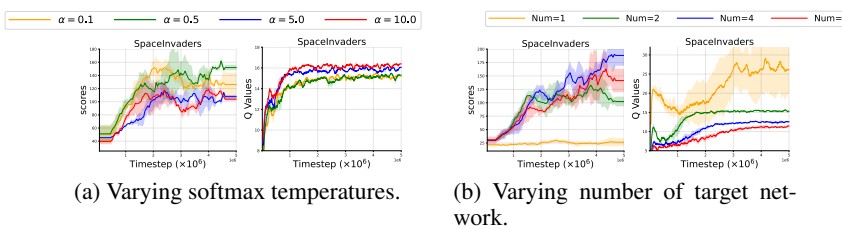

(a) Varying softmax temperatures.

(b) Varying number of target network.

Figure 6: (a) Performance (left) and Q values (right) of Highway DQN using varying softmax temperatures $\alpha$. (b) Corresponding results using varying numbers of target networks. Average over 5 seeds, 1 standard deviation.

## C.4 ADDITIOANL RESULTS ON ABLATION STUDY

We evaluate our Highway DQN with varying softmax temperature $\alpha$ and number of target networks. As shown in Fig. 6a and Fig. 6b, appropriate softmax temperature $\alpha$ and number of target networks can reduce the overestimation of $Q$ values and benefit the performance of the algorithm.

