# OpenReview forum: "Highway Reinforcement Learning"
_ICLR.cc/2023/Conference — Submitted to ICLR 2023_

### Official Review · Reviewer_dbJn · 2022-10-22

**Confidence:** 4
**Correctness:** 4
**Technical Novelty And Significance:** 3
**Empirical Novelty And Significance:** 2
**Recommendation:** 6

**Clarity, Quality, Novelty And Reproducibility:**

The paper is well-written and clear.

The proofs seem to be correct.

The algorithm seems to be novel and interesting, as explained above.

The code is available for reproducibility purposes, but I did not attempt to run it.

**Strength And Weaknesses:**

### Strengths:

1. The paper presents an interesting and novel idea that is potentially impactful in the off-policy RL community. It is a longstanding goal to be able to learn off-policy in a theoretically well-motivated way without needing high-variance importance weights. Hopefully the paper could spur follow-up work that tightens up the main ideas in the model-free case.
2. The proofs seem to be correct, while limited to the model-based setting where the model of the environment is known to allow for exact computation of the relevant expectations.
3. While the experiments are somewhat small-scale, they present consistent evidence of improvement, especially in the model-based case where the theory actually holds.


### Weaknesses:

1. Important baselines are missing from the experiments. First, there is no comparison to any importance-weighted off-policy algorithm like retrace. Such a baseline is important to actually substantiate the claims made in the paper about the benefit of avoiding importance sampling. Second, it seems from my reading that the Highway DQN algorithm uses maxmin DQN but the multi-step DQN baseline does not. The number of Q functions is important, see fig. 5b, so this seems to be an important issue. If this is the case, then a missing baseline is the multi-step DQN with maxmin learning on top to reduce overestimation and make a fair comparison to the implementation of highway DQN (if I have misunderstood how the implementation works, let me know).
2. There is little to no discussion of the tradeoffs and potential weaknesses of the algorithm. In particular, I am somewhat concerned about what happens to the algorithm in the model-free case when rather than taking expectations of n-step returns under each of the candidate policies, we instead only see the one trajectory that already exists in the replay buffer. Especially in stochastic environments, maximizing over these single-sample returns seems like it may cause worse than usual overestimation problems and high variance.

**Summary Of The Paper:**

This paper presents highway RL, a novel dynamic programming-based strategy for off-policy RL that incorporates multistep information without requiring importance weights. Highway RL replaces the traditional Bellman backup by the maximum over both $n$ and $\pi$ of the $n$-step return of $\pi$ followed by the maximum over the learned value function (where $\pi$ is taken from some candidate set of policies $\hat \Pi$). The algorithm is proved to converge in the model-based setting and it is shown that the rate can be better than standard VI under an assumption that the candidate policy set covers the optimal policy. The algorithm is then extended heuristically to the model-free setting. Experiments using the model-based algorithm in some toy problems and the model-free algorithm on MinAtar demonstrate performance better than the chosen baselines.

**Summary Of The Review:**

Overall, I liked this paper and think it presents an interesting and novel algorithm. There are still a few issues regarding baselines and discussions of tradeoffs, but these do not seem like fatal flaws, so I rate it as a weak accept.

---

> ### Author Response · Authors · 2022-11-15
> **Response to Reviewer dbJn**
>
>
> Thanks for the valuable comments and advice. Below we give detailed responses to the reviewer's questions.
>
>
>
> ###  Q1:
>
> >  1. ... First, there is no comparison to any importance-weighted off-policy algorithm like retrace... Second, it seems from my reading that the Highway DQN algorithm uses maxmin DQN but the multi-step DQN baseline does not. The number of Q functions is important, see fig. 5b, so this seems to be an important issue...
>
> ###  A1:
>
> Thanks for these very helpful comments.
>
> Following the reviewer's suggestions, we implemented Retrace($\lambda$) on top of Maxmin DQN. We set the hyperparameter $\lambda=1$ as suggested by the Retrace paper.  The best number of target networks was chosen from $\\{2, 4, 6\\}$, and the best lookahead step was chosen from $\\{4, 8, 12\\}$, which follows exactly the same setting as Highway DQN. We added the results using $5$ random seeds in the updated Fig. 5. For the reviewer's convenience, we report the mean and the standard deviation of the scores averaged over the last 10% training steps and 5 random seeds. As shown in the Table below, our Highway DQN significantly outperforms Retrace on 3 of 5 tasks while performing on par with it on the remaining 2 tasks.
>
>
> |               | Highway DQN         | Retrace            |
> | ------------- | ------------------- | ------------------ |
> | Asterix       | **37.9 $\pm$ 3.6**  | 25.7 $\pm$ 1.7     |
> | Breakout      | 20.4 $\pm$ 2.1      | **20.5 $\pm$ 1.8** |
> | SpaceInvaders | **180.6 $\pm$ 9.1** | 139.2 $\pm$ 10.2   |
> | Seaquest      | **24.7 $\pm$ 3.2**  | 19.57 $\pm$ 7.9    |
> | Freeway       | 62.4  $\pm$ 0.7     | **64.5 $\pm$ 1.2** |
>
>
>
>
> Regarding the second concern of the reviewer, our multi-step DQN is also implemented on top of Maxmin DQN. These details were initially in APPENDIX Sec C.3.2 due to space constraints. We added more implementation details to the new main part of the paper.
>
>
>
>
>
> ###  Q2:
>
> >  2. There is little to no discussion of the tradeoffs and potential weaknesses of the algorithm... Especially in stochastic environments, maximizing over these single-sample returns seems like it may cause worse than usual overestimation problems and high variance.
>
> ###  A2:
>
> Thanks for this insightful comment.
> The issue of deciding the number of trial trajectories for each policy also exists in classical RL algorithms, e.g., Multi-Step SARSA, and Monte Carlo. While more trials per policy may improve the estimation accuracy, they may also cost more samples and reduce sample efficiency.
> On the other hand, while a larger number of behavioral policies or lookahead steps may increase efficiency, it  might incur overestimation issues when the estimate is biased, leading to high variance.
>
> In our implementation, we use softmax and Maxmin DQN to alleviate the overestimation issue caused by insufficient sampling. We included a more detailed discussion of the tradeoffs and potential weakness in the revised paper.

---

### Official Review · Reviewer_NWV2 · 2022-10-23

**Confidence:** 4
**Correctness:** 3
**Technical Novelty And Significance:** 3
**Empirical Novelty And Significance:** 1
**Recommendation:** 5

**Clarity, Quality, Novelty And Reproducibility:**

Yes, it's clear. The paper's idea is original. Reproducibility: it's unclear from the paper, as the authors did not submit the code.

**Strength And Weaknesses:**

Strength:
- The entire paper is clearly written and easy to follow.

-  The idea of highway RL is quite natural. The authors also provide theories for justification of the algorithm, though the derivations seem mostly to follow from the prior works on the one-step Bellman operator.

Weakness:
-  The computation of the proposed algorithm is very expensive for each step, though it can achieve an improved convergence rate. It needs to search over the product space of possible policies $\hat{\Pi}$ and possible lookahead steps $\mathcal{N}$, which can be very large. Moreover, the algorithm is also much more complicated than the standard Bellman operator..

-  Theorem 5 shows that the proposed operator can achieve a convergence rate of $\gamma^N$. However, if we apply the original Bellman operator for $N$ times for each time step, i.e., $B^{\circ N}$, we should also achieve a convergence rate of $\gamma^N$, right? If this is true, I am not sure what's the advantage of your operator over the $B^{\circ N}$ operator. Let alone the per-step computational cost for your operator is $O((|A|+|\Pi||\mathcal{N}|)|S|^2)$ , and  $O(N|A||S|^2)$ for $(B)^{\circ N}$.


-  Can you provide a comparison with the method from He et al., 2017? This work is quite relevant, as they also proposed an algorithm for improving the convergence of Q-learning by looking ahead. The high-level idea is that they build the lower and upper bound of the Q function, and make them two constraints. So, when doing the Bellman update, it needs to solve a constrained optimization problem, which can be efficiently solved using the Lagrange multiplier trick.



-  Abstract: "We first derive a new multistep Value Iteration (VI) method that converges to the optimal Value Function (VF) with an exponential contraction rate but linear computational complexity".
Linear computational complexity in what? It would be better to state it more explicitly in the abstract, though it can be inferred from the later sections.

- As you mentioned in the introduction (page 2),
> "It effectively assigns credit over multiple time steps",

  I feel the term "credit assignment" is a bit vague. What do you mean by assigning credit over multiple time steps? Can you provide a more formal explanation of this?

- Can you explain more formally why your operator doesn't suffer from high variance?

- For equations (9) & (11), why do you need to take a max over {0, n'}?

- Do you have any results on Atari games? The MinAtar Games is not a very commonly adopted benchmark. It would be much more accessible for the readers if you have comparisons on more commonly used benchmarks.

- Page 7: I don't understand why $(B)^{\circ N}$ will incur a greater computational cost. For your algorithm, each step will suffer from a computational cost of $O((|A|+|\Pi||\mathcal{N}|)|S|^2)$ cost, while for $(B)^{\circ N}$, the cost should only be $O(N|A||S|^2)$, which is smaller than yours.

- Won't $|M_{s_0, a_0}|$ become very large for RL tasks, as the policy will be updated frequently in online learning? This will also increase the computational cost for each update step.

 - Figure 4 (a): it seems that increasing the number of steps doesn't improve the performance, and it even makes the performance worse. Can you explain why?


======
Post-rebuttal:
Thanks for the authors for answering my questions. I increase the rating to 5.

[1] He, Frank S., Yang Liu, Alexander G. Schwing, and Jian Peng. "Learning to play in a day: Faster deep reinforcement learning by optimality tightening."  ICLR 2017.

**Summary Of The Paper:**

This paper proposes highway RL, which adaptively selects the policy and look-ahead steps for the Bellman operator. The proposed algorithm doesn't need off-policy corrections and hence doesn't suffer from the large variance. In addition, it can also achieve an improved convergence rate than the conventional one-step Bellman operator. The authors also present the application of Q-learning for both continuous and discrete domains. Empirically, the proposed variant can attain better results than the baseline methods, including Multi-step DQN, Maxmin DQN, and Multi-step SARSA on a series of RL benchmarks.


**Summary Of The Review:**

Overall, I think the authors propose an interesting and theoretically justified idea for improving the convergence rate of the Bellman operator. The idea itself is novel. However, the new operator seems doesn't improve the convergence rate, and the per-step computational cost $O((|A|+|\Pi||\mathcal{N}|)|S|^2)$ is pretty large. So, I am not convinced about the improvement of the proposed operator over the standard Bellman operator or  $B^{\circ N}$. I am happy to increase the rating if the authors can address my concerns raised in the strength and weakness section well. At the current stage, I recommend for rejection.

---

> ### Author Response · Authors · 2022-11-15
> **Response to Reviewer NWV2 (2/2)**
>
>
> ###  Q3:
>
> >  Can you provide a comparison with the method from He et al., 2017?....
>
> ###  A3:
>
>
> He et al. (2017) also use the idea of searching across various lookahead steps, but they do not search over various policies. They employ the lookahead returns to construct additional inequality bounds on the VF. Our work instead derives a novel Bellman operator on updating the VF directly and provides a thorough theoretical analysis of the convergence properties.
> Another difference is that we propose a novel, theoretically justified softmax operation instead of the max operation to alleviate the potential overestimation issue.
> We added the discussion above to the related work.
>
>
> ###  Q4:
>
> >  Can you explain more formally why your operator doesn't suffer from high variance?
>
> ###  A4:
>
> Our operator does *not* involve importance sampling for correcting off-policy data correction (as we illustrated in Theorem 1, 2), therefore it does not suffer from the high variance caused by the product of the importance sampling ratios.
> However, there exists potential variance due to the overestimation issue caused by the max or softmax.
> More research is needed to understand how the variance of our method compares to that of advanced IS-based variance reduction methods like Retrace($\lambda$).
>
>
>
>
> ###  Q5:
>
> >  For equations (9) and (11), why do you need to take a max over $\{0, n'\}$?
>
> ###  A5:
>
> The operation $\max_{ n \in \{0, n'\}}$ is the key to guarantee the unbiased property w.r.t. $V^*$ when we use softmax operation $\underset{\pi \in \widehat{\Pi} }{\mathop{smax}^\alpha} \underset{n \in \mathcal{N} }{\mathop{smax}^\alpha}$ (the softmax is necessary for alleviating the overestimation issue).
> Without $\max_{ n \in \{0, n'\}}$ in eq. (9), the operator will be biased w.r.t. $V^*$.
> We provided some discussion below Theorem 2. And we added more detail in the updated APPENDIX A.3.
>
>
>
>
>
> ###  Q6:
>
> >  Do you have any results on Atari games? The MinAtar Games is not a very commonly adopted benchmark. It would be much more accessible for the readers if you have comparisons on more commonly used benchmarks.
>
> ###  A6:
>
> Thanks for this suggestion. While our paper focuses on providing a powerful theory of off-policy learning, it would be interesting to evaluate our algorithm on Atari games in future work. We chose the MinAtar suite because while it does not require the prohibitive expense of the ALE benchmark, it still includes important RL challenges such as reward sparsity and visual states. Note that thanks to these properties, these environments have been increasingly adopted by the RL community.
>
>
>
>
>
> ###  Q7:
>
> >  Figure 4 (a): it seems that increasing the number of steps doesn't improve the performance, and it even makes the performance worse. Can you explain why?
>
> ###  A7:
>
> Multi-step DQN is very sensitive to the number of lookahead steps (see Figure 4 (a) (right)). This is because Multi-step DQN uses
>
> $$G_t^n=\sum_{n=0}^{N-1} \gamma^n r_{t+n} + \gamma^N  \max_{a_{t+N}} Q(s_{t+N},a_{t+N}),$$
>
> which is more likely to be affected by the quality of the data within $N$ steps as $N$ increases. Since the replay buffer often contains bad data, a large $N$ could damage the performance of the algorithm.
>
> In contrast, our Highway DQN shows robustness against variations of the lookahead step (see Figure 4 (a) (left)). This is because by taking (soft) maximization over the $n$-step returns, i.e., $\underset{n \in \mathcal{N} }{\mathop{smax}^\alpha} \max\limits_{n' \in \{0, n\}} G_t^{n'}$, our method can adaptively decide the weight on each $n$-step return for each lookahead step $n$, making it less sensitive to the hyperparameter of lookahead steps.
>
>
>
>
>
> ###  Q8:
>
> >  As you mentioned in the introduction (page 2),"It effectively assigns credit over multiple time steps" ... What do you mean by assigning credit over multiple time steps?...
>
> ###  A8:
>
> By "it effectively assigns credit over multiple time steps" we mean that "our operator can directly assign the future credit to the past states across multiple time steps." We illustrated this statement in Figure 1 (right), where the credit information at the end can be quickly transported to past actions. We clarified this point on page 2 in the revised version.
>
>
>
> ###  Q9:
>
> >  Abstract...Linear computational complexity in what? It would be better to state it more explicitly in the abstract, though it can be inferred from the later sections.
>
> ###  A9:
>
> Thanks for this helpful comment.
> Our operator's computational complexity is linear in the number of behavioral policies and lookahead steps. We rewrote this statement in the revised abstract.

---

> ### Author Response · Authors · 2022-11-15
> **Response to Reviewer NWV2 (1/2)**
>
>
> We sincerely appreciate reviewer NWV2's constructive comments.
> Below we address the questions raised by the reviewer.
>
>
>
> ###  Q1:
>
> >  - The computation of the proposed algorithm is very expensive for each step, though it can achieve an improved convergence rate. It needs to search over the product space of possible policies $\widehat{\Pi}$ and possibly lookahead steps $\mathcal{N}$, which can be very large.
> > - Won't $|M_{s_0,a_0}|$ become very large for RL tasks, as the policy will be updated frequently in online learning?...
> > - Moreover, the algorithm is also much more complicated than the standard Bellman operator.
>
> ###  A1:
>
> The computational complexity of our method is reduced by limiting the search space without losing theoretical soundness. Moreover, the computation can be greatly sped up  by parallelization.
> Please find an illustration below on how these techniques are implemented in our Highway Algorithms.
>
> - For our Highway VI, we limit the search scope by $|\widehat{\Pi}|=5$ and $|\mathcal{N}|=10$.
>   Note that as we explain in A2 below, this choice corresponds to roughly the same computational cost as applying Bellman Operators for $10$ times (where $|\mathcal{A}|=4$).
>   Moreover, the computation for each policy and lookahead is calculated in parallel on GPU.
>   In multi-room environments, our Highway Value Iteration requires only half the runtime of classical Value Iteration on various tasks (see Fig. 2 (a) (right) in the paper).
> - In our implementation of Highway DQN, we search over a random subset of $M_{s_0,a_0}$ - this is theoretically justified because our operator is unbiased for any set of behavioral policies  (see Theorem 1 and 2). Compared to DQN, Highway DQN costs *1.5 times* more runtime but achieves *twice* the score on 3 of 5 tasks and *thrice* the score on the Space Invader task.
> - Regarding the implementation, our method does *not* require quite a lot of modification of existing algorithms like DQN. The new thing is just to save the data collected by policy $\pi_m$ into the corresponding $m$-indexed replay buffer and compute the Q target according to the well-defined formula in eq. (11).
>
>
> We clarified the above points in the new Algorithms B.3 and B.4.
>
>
>
> ###  Q2:
>
> >  - Theorem 5 shows that the proposed operator can achieve a convergence rate of $\gamma^N$. However, if we apply the original Bellman operator for $N$ times for each time step, i.e.,  $(B)^{\circ N}$, we should also achieve a convergence rate of $\gamma^N$, right? If this is true, I am not sure what's the advantage of your operator over the $B^{\circ N}$ operator.
> >  - Page 7: I don't understand why $(B)^{\circ N}$ will incur a greater computational cost. For your algorithm, each step will suffer from a computational cost of $O((|A|+|\mathcal{N}||\widehat{\Pi}|)|\mathcal{S}|^2)$ cost, while for $(B)^{\circ N}$, the cost should only be $O(N|\mathcal{A}||\mathcal{S}|^2)$, which is smaller than yours.
>
> ###  A2:
>
> Thanks for this critical comment.
> We claim that $(\mathcal{B})^{\circ N}$ will incur a greater computational cost than ours in the *model-free* case but *not* in the *model-based* case.
> Note that the major difference is that our Highway operator $\max_{\pi \in \widehat{\Pi }} \max_{ n \in \mathcal{N} } \left( \mathcal{B} ^{\pi} \right) ^{\circ n}\mathcal{B}$ involves applying  $\mathcal{B}^{\pi}$ instead of $\mathcal{B}$ for several times.
>
>
> First, we would like to clarify that in the *model-based* case, the complexities (ours: $O((|A|+|\mathcal{N}||\widehat{\Pi}|)|\mathcal{S}|^2)$; $(\mathcal{B})^{\circ N}$: $O(N|\mathcal{A}||\mathcal{S}|^2)$) are very close when we set $|\mathcal{N}|\approx N$ and $|\widehat{\Pi}|\approx |{\mathcal{A}}|$ (exactly what we use in practice); and ours is smaller when the action space is very large, and the number of behavioral policies is manually limited, i.e., $|\widehat{\Pi}| \ll |{\mathcal{A}}|$.
>
> Second, in the *model-free* case, our operator, which involves
> $\left( \mathcal{B} ^{\pi} \right)^{\circ n}  \mathcal{B}$, shows its advantage by utilizing the $n$-step trajectory data generated by $\pi$ with minor cost (just by accumulating rewards within $n$-step, i.e., $\sum_{n'=0}^n{\gamma^{n'}} r_{t+n'}+\gamma^n \max Q(s_{t+n}, a_{t+n})$).
> While $(\mathcal{B})^{\circ N}$ can only utilize the $1$-step data and needs to update the value function for $N$ times.
> When using an approximated value function, $\mathcal{B}^{\circ N}$ must optimize the following objective for $N$ times
> $$ \theta_{k+1} \leftarrow \min_{\theta}
> \left[ Q_{\theta}(s_{t}, a_{t}) -
>   r_t - \gamma \max_{a_{t+1}} Q_{\theta_{k}}(s_{t+1}, a_{t+1})  \right]^2, $$
> which implies a greater computational cost.
>
> We added more explanations in Sec 5 to clarify these points.

---

### Official Review · Reviewer_7nDd · 2022-10-24

**Confidence:** 3
**Correctness:** 3
**Technical Novelty And Significance:** 3
**Empirical Novelty And Significance:** 3
**Recommendation:** 6

**Clarity, Quality, Novelty And Reproducibility:**

The paper is written clearly and is of good quality. The proposed operators and algorithms are novel. GitHub links are included to reproduce the numerical results.

**Strength And Weaknesses:**

As far as I can see, the advances of the proposal include:

1. the introduction of the highway Bellman-operator;
2. the development of the associated RL algorithms built upon the Bellman operator;
3. the established theoretical results accompanying the Bellman operator;
4. empirical verifications of the highway algorithm.

Below are some of the points to discuss:

1. The proposed Bellman operator has very nice theoretical properties. Nonetheless, it seems to involve several hyper-parameters, including the set of the policies as well as the choice of the maximal lookahead steps. It remains unclear to me whether the operator and the associated algorithms are sensitive to the set of policies. How would recommend to set these hyper-parameters in practice?
2. The theories are mainly concerned with the contraction properties of the Bellman operator. Could you please derive some theoretical results for certain highway algorithms?

**Summary Of The Paper:**

The paper is concerned with solving general reinforcement learning problems under the MDP formulation. The author(s) introduced a highway Bellman operator that enjoys nice theoretical properties when compared to standard one-step or multi-step Bellman operators. Built upon this Bellman operator, three RL algorithms are developed. Numerical studies were conducted to support the claims.

**Summary Of The Review:**

The paper develops a class of highway reinforcement learning algorithms based on a highway Bellman operator. The methods are novel in my opinion. The proposal is justified via theoretical results and numerical experiments.

---

> ### Author Response · Authors · 2022-11-15
> **Response to Reviewer 7nDd**
>
>
> Thanks for the Reviewer 7nDd's insightful and helpful comments.
> Please see our response below.
>
>
>
> ###  Q1:
>
> >  1. ...it seems to involve several hyper-parameters, including the set of policies as well as the choice of the maximal lookahead steps. It remains unclear to me whether the operator and the associated algorithms are sensitive to the set of policies. How would recommend to set these hyper-parameters in practice?
>
> ###  A1:
>
>
> In general, the best choice of number of behavioral policies and lookahead steps will depend on the environment.
> A larger number of behavioral policies $\widehat \Pi$ and maximal lookahead steps $N$ usually leads to better performance, at the expense of a greater computational cost and the risk of overestimation.
> We suggest use as heuristic $|\widehat \Pi| \approx |\cal A|$, such that the behaviroal policies can potentially cover the action space, and $N\leq 20$.
> Please find more details about the hyperparameters used in APPENDIX C.
>
>
> In order to test the sensitivity of our method with respect to the number of policies in the buffer, we run an experiment while changing the policy buffer size. As listed in the table below, our Highway VI achieves the best performance when the number of policies exceeds $10$.
>
>
>
> | Size of the buffer of behavioral policies | 1    | 5    | 10   | 15   |
> | ----------------------------------------- | ---- | ---- | ---- | ---- |
> | Required iterations to converge           | 97.2 | 62.1 | 57.3 | 57.2 |
>
>
>
>
>
>
>
> ###  Q2:
>
> >  2. ...Could you please derive some theoretical results for certain highway algorithms?
>
> ###  A2:
>
> We thank reviewer 7nDd for raising this constructive comment.
> For the Highway Value Iteration Algorithm (Algorithm A.1), we provide a preliminary convergence rate of $\gamma$ for dynamically-changing behavioral policies (Theorem 1 Point 3).
> It would also be interesting to study the behavior of our algorithm under function approximation with finite-sample. We leave these problems for future work.

---

### Official Review · Reviewer_ftfH · 2022-10-29

**Confidence:** 4
**Correctness:** 2
**Technical Novelty And Significance:** 2
**Empirical Novelty And Significance:** 3
**Recommendation:** 5

**Clarity, Quality, Novelty And Reproducibility:**

The paper is written in a clear way. I think the idea itself seems novel and gives interesting numerical results but at this point, it is more of a heuristic and is not well-supported by the theoretical results, which are lacking/misleading.

**Strength And Weaknesses:**

Strengths:
- The idea of using suboptimal policies that might need to better credit assignment is an interesting one.
- The numerical results seem to be promising on a set of difficult problems.

Weaknesses:
- [Theorem 1] The fact that the new operator is a contraction eventually relies on the single step of B. I feel that it is misleading to state the ||GV - GV'|| <= ||BV - BV'||. In the proof of theorem 1, the fact that 0 \in N is explicitly used (which, in effect, ignores all of the behavioral policies) to show the \gamma-contraction property and the fixed point property.
- [Theorem 5] This result was also underwhelming in that it is basically doing multi-step optimal Bellman operators. The assumption is stated in a way that makes it slightly weaker in that it requires one of the behavioral policies to be equivalent to the optimal policy at a subset of states.
- A few other papers that use multi-step operators are:
"Online Planning with Lookahead Policies"
"Feedback-Based Tree Search for Reinforcement Learning"
"Multiple-Step Greedy Policies in Online and Approximate Reinforcement Learning"

**Summary Of The Paper:**

The paper suggests a new way of doing multi-step RL using a Bellman operator that first performs a few steps of rollout via behavioral policies, and then doing one step of the optimal Bellman operator. The authors term this new method Highway RL. The paper first shows theoretical results (in an exact DP setting) and then shows how the new operator can be used in VI, Q learning, and DQN. Lastly, they show empirical results.

**Summary Of The Review:**

It would be nice to see if the authors could take their idea and do a deeper analysis of how the operator behaves as a function of the quality of the behavioral policies.

---

> ### Author Response · Authors · 2022-11-15
> **Response to Reviewer ftfH**
>
>
> We sincerely thank reviewer ftfH for the valuable comments.
> Below we address the questions raised by the reviewer.
>
>
>
> ###  Q1:
>
> > [Theorem 1] ...I feel that it is misleading to state the $\| $${\mathcal{G}}_{\mathcal{N}}^{\widehat \Pi}V-{\mathcal{G}}\_{\mathcal{N}}^{\widehat \Pi} V'\|\leq\|{\mathcal{B}}V-{\mathcal{B}}V'\|$$$. In the proof of theorem 1, the fact that $0 \in \mathcal{N}$ is explicitly used (which, in effect, ignores all of the behavioral policies) to show the $\gamma$-contraction property and the fixed point property.
>
> ###  A1:
>
> Thanks for this critical comment.
> We would like to clarify that $0 \in \mathcal{N}$ is necessary for the fixed point property but *not* for the contraction property.
> In fact, if $0 \notin \mathcal{N}$, our operator can obtain better contraction property:
> $$\|{\mathcal{G}}_{\mathcal{N}}^{\widehat \Pi} V-{\mathcal{G}}\_{\mathcal{N}}^{\widehat \Pi} V' \| \le \max\_{\pi \in \widehat{\Pi }} \max\_{n \in \mathcal{N}} \gamma ^n\left\| \mathcal{B} V-\mathcal{B} V' \right\| \leq \gamma \| \mathcal{B} V - \mathcal{B} V' \| \leq \gamma^2 \|  V - V' \|
> $$
> The proof is similar to the one of Theorem 1 Point 1.
> We clarified this point below Theorem 1 in the revised paper to avoid misunderstandings.
>
> ###  Q2:
>
> >  [Theorem 5] This result was also underwhelming in that it is basically doing multi-step optimal Bellman operators. The assumption is stated in a way that makes it slightly weaker in that it requires one of the behavioral policies to be equivalent to the optimal policy at a subset of states.
>
> ###  A2:
>
> We would like to clarify a possible misunderstanding: the condition in Theorem 5 only requires that for each state in the MDP, the set of behavioral policies contains a policy that acts optimally in some "neighborhood" of that state (and could be defined arbitrarily in the rest of the state space).
> Note that this condition is much weaker than having some optimal or near-optimal policies included in the behavioral policy set.
>
> Note that although it is not surprising that a multi-step operator has a convergence rate of $\gamma^N$, some operators might not have $V^*$ as the fixed point when using arbitrary off-policy data.
> As stated in Table 1, for the widely-used Multi-step Bellman Optimality Operator, unbiased learning only happens when *all* the behavioral policies are optimal (the proof is provided in Sec A.2).
>
>
>
>
>
>
>
> ###  Q3:
>
> > It would be nice to see if the authors could take their idea and do a deeper analysis of how the operator behaves as a function of the quality of the behavioral policies.
>
> ###  A3:
>
> Thanks for this insightful comment. We made such an analysis in Theorem 3 and 5, showing our operator can accelerate the convergence when one of the behavioral policies acts well over $N$ steps.
>
> To further address the concerns of the reviewer, we ran an experiment on a Multi-Room environment to show how our operator performs as a function of the quality of the behavioral policies. As listed in the table below, our operator requires fewer iterations to converge as the quality of the behavioral policies increases.
>
>
>
> | Quality of Behavioral Policies  | Good         | Medium       | Bad           | Value Iteration (No Behavioral Policies) |
> | ------------------------------- | ------------ | ------------ | ------------- | ---------------------------------------- |
> | Required Iterations to converge | 39 $\pm$ 2.1 | 97 $\pm$ 8.7 | 190 $\pm$ 9.2 | 270 $\pm$ 0.2                            |
>
>
>
>
> (The experiments were conducted in a 30-rooms environment. We collected $30$ policies generated by value iteration during the training process and divided them into 3 groups ordered by their scores. Then we run Highway Value Iteration with these fixed behavioral policy sets.)
>
>
>
> ###  Q4:
>
> > A few other papers that use multi-step operators are: "Online Planning with Lookahead Policies" "Feedback-Based Tree Search for Reinforcement Learning" "Multiple-Step Greedy Policies in Online and Approximate Reinforcement Learning"
>
> ###  A4:
>
>
> We thank the reviewer for pointing out the related work [1, 2, 3]. We discussed the relations and differences between these methods and our method with these works and cited them in the revised version.
>
> Efroni et al. (2018, 2020) and Jiang et al. (2018) also adopt the idea of taking the most promising values by searching over various policies and lookahead steps [1, 2, 3].
> However, these works need to search over the product of the original policy space or the action space for multiple lookahead steps, relying on additional solvers for the search problem.
> In contrast, our method needs to search over only a limited set of behavioral policies with minor costs.
>
> [1] Efroni et al. (2018). Multiple-step greedy policies in approximate and online reinforcement learning. NeurIPS.
>
> [2] Efroni et al. (2020). Online planning with lookahead policies. NeurIPS.
>
> [3] Jiang et al. (2018). Feedback-Based Tree Search for Reinforcement Learning. ICML

---

> > ### Comment · Reviewer_ftfH · 2022-11-22
> > **Response**
> >
> > Thank you for the responses! A few comments:
> >
> > Regarding A1, thanks for the clarification. In my mind, if the fixed point is not "correct", then the fact that contraction holds does not have too much meaning in the context of MDPs or RL. Therefore I think 0 \in N being required essentially makes the new approach fall back to the standard theory.
> >
> > For A3, I would be curious to see the performance as a function of total computation used, including the time needed to find the good behavioral policies. If we look at this way, is there still a benefit to the new approach?

---

> > > ### Author Response · Authors · 2022-11-26
> > > **Further response to reviewer ftfH**
> > >
> > > We thank the reviewer for the constructive feedback.
> > >
> > > ### Q:
> > >
> > > > Regarding A1 ... Therefore I think $0 \in N$ being required essentially makes the new approach fall back to the standard theory.
> > >
> > > ### A:
> > >
> > > Regarding A1, $0\in \mathcal{N}$ does *not* mean our theory falls back to the standard theory of the Bellman Optimality (BO) Operator.
> > > Note that our operator is equivalent to BO operator *only* when ${\mathop{argmax}}_{n \in \mathcal{N}} ( \mathcal{B}^\pi)^{\circ n} \mathcal{B} V(s) =0$ holds for any $s \in \mathcal{S}$.
> > > However, this is *not* the general case in practice.As shown in the table below, we have more than 98% of the samples satisfying ${\mathop{argmax}}\_{n \in \mathcal{N}}( \cdot )>0$ during the training process.
> > >
> > > Besides, the standard theory of the BO Operator does *not* share some good properties of our Highway Operator.
> > > We theoretically show that in the cases when $\mathop{argmax}_{n \in \mathcal{N}}( \cdot ) > 0$, our operator converges faster than the BO operator (see Theorem 3). Moreover, our operator has a convergence rate of $\gamma^N$ under some mild conditions (see Theorem 5).
> > >
> > > | $\mathop{argmax}_{n \in \mathcal{N}}( \cdot )$ | 0    | 1     | 2     | $> 2$ |
> > > | ---------------------------------------------- | ---- | ----- | ----- | ----- |
> > > | Percentage                                     | 1.2% | 17.1% | 19.5% | 62.2% |
> > >
> > > (The table above lists the statistics of $\mathop{argmax}_{n \in \mathcal{N}}(\mathcal{B}^\pi)^{\circ n}\mathcal{B} V (s)$ during the training process of Highway VI in the multi-room environment with 30 rooms.)
> > >
> > >
> > >
> > > ### Q:
> > >
> > > > For A3, I would be curious to see the performance as a function of total computation used, including the time needed to find the good behavioral policies. If we look at this way, is there still a benefit to the new approach?
> > >
> > > ### A:
> > >
> > > Regarding this suggestion, we show the runtime of our Highway VI and VI.
> > > As listed in the table below, Highway VI requires much less runtime than VI to converge to the optimal value function.
> > >
> > > | Runtime (Clock Time) | Error of Highway VI           | Error of VI        |
> > > | -------------------- | ----------------------------- | ------------------ |
> > > | 400                  | 823                           | **729**            |
> > > | 800                  | 570                           | **510**            |
> > > | 1200                 | **0.8**                       | 328                |
> > > | 1600                 | $ \mathbf{ 3\times10^{-9} }$  | $7.9$              |
> > > | 2000                 | $ \mathbf{ 4\times10^{-13} }$ | $6\times 10^{-13}$ |
> > >
> > > Error is defined as $|| V_{k}-V^*  ||_{\infty}$.
> > > Note that the time for finding the behavioral policies is also included for our Highway VI.
> > > (Here are the experimental details.We conduct experiments in a multi-room environment with 90 rooms.The behavioral policies are generated by the learned value functions in the learning process (see Algorithm 1). This is different from the experiment in A3, which generated the behavioral policies by some other algorithms, aiming to do an ablation study to address the concern of "how the operator behaves as a function of the quality of the behavioral policies.")
> > >
> > >
> > > ---
> > >
> > > We would be glad to engage in discussions if reviewer ftfH has any further concerns.

---

> > > ### Author Response · Authors · 2022-12-08
> > > **Was our response satisfactory?**
> > >
> > > Thanks to Reviewer ftfH for valuable feedback and comments. We have updated our replies and added experimental results as suggested. Since the discussion phase will close soon, could you please confirm that all concerns have been addressed adequately?

---

### Decision · Program_Chairs · 2023-01-20

**Decision:**

Reject

**Justification For Why Not Higher Score:**

Multiple reviewers remained unconvinced on the theoretical results.  Reviewers also asked for more discussions around the sensitivity of the algorithm regarding the added hyper-parameters, chosen behavior policy set,  and the added complexity in each update to search over the product of behavior policy set and look-ahead steps. Comparisons to more recent off-policy RL algorithms are missing.

**Justification For Why Not Lower Score:**

N/A

**Metareview: Summary, Strengths And Weaknesses:**

The paper proposed a new highway Bellman operator for multi-step RL which adaptively selects the behavior policy for rollout and look-ahead steps for the Bellman operator. Theoretical results suggest highway Bellman operator enjoys an improved convergence rate than the conventional one-step Bellman operator. The authors also showed that empirically highway RL performed better than some of the baselines. Multiple reviewers remained unconvinced on the theoretical results.  Reviewers also asked for more discussions around the sensitivity of the algorithm regarding the added hyper-parameters, chosen behavior policy set,  and the added complexity in each update to search over the product of behavior policy set and look-ahead steps. Comparisons to more recent off-policy RL algorithms are missing.